# Active Set Ordering

**Quoc Phong Nguyen[1,3], Sunil Gupta[1],**
**Svetha Venkatesh[1], Bryan Kian Hsiang Low[2], Patrick Jaillet[3]**
[1]Applied Artificial Intelligence Institute, Deakin University, Australia
[2]School of Computing, National University of Singapore, Singapore
[3]LIDS and EECS, Massachusetts Institute of Technology, USA
`qphongmp@gmail.com, sunil.gupta@deakin.edu.au,`
`svetha.venkatesh@deakin.edu.au, lowkh@comp.nus.edu.sg, jaillet@mit.edu`

## Abstract

In this paper, we formalize the active set ordering problem, which involves actively discovering a set of inputs based on their orderings determined by expensive evaluations of a blackbox function. We then propose the *mean prediction* (MP) algorithm and theoretically analyze it in terms of the regret of predicted pairwise orderings between inputs. Notably, as a special case of this framework, we can cast *Bayesian optimization* as an active set ordering problem by recognizing that maximizers can be identified solely by comparison rather than by precisely estimating the function evaluations. As a result, we are able to construct the popular *Gaussian process upper confidence bound* (GP-UCB) algorithm through the lens of ordering with several nuanced insights. We empirically validate the performance of our proposed solution using various synthetic functions and real-world datasets.

## 1 Introduction

In real-world applications, we often encounter the problem of estimating an unknown function, known as a *blackbox function*, (i.e., those without closed-form expressions or derivatives) using their expensive and noisy evaluations. Under these circumstances, an efficient sequential process of evaluating the function is desired. On one extreme, *experimental design* (ED) aims to estimate the function in its entire input domain, e.g., by decreasing the uncertainty of the function globally in Bayesian ED [4, 18]. On the other extreme, the renowned *Bayesian optimization* (BO) targets inputs with the extreme function values such as the maximizers and the minimizers [2, 6, 7].

While the connection between ED and BO is studied in the classic work of [19], we still lack a problem formulation that strikes a balance between the prohibitively expensive process of estimating the entire function globally in ED and the lack of information about the function away from extreme locations in BO. One may consider a related problem, called *level set estimation* (LSE), which focuses on estimating inputs with function evaluations above or below a given (or implicit) threshold [1, 3, 8, 14]. However, without domain knowledge of the blackbox function, it is easy to set a threshold that leads to undesirably large or small level sets.

Let us consider an environmental monitoring problem of estimating a chemical concentration in a field. The blackbox function is the mapping from locations of the field to the chemical concentration measurement. It can be of a greater interest to estimate the maximizers, the minimizers, the top-$k$ locations (with the highest chemical concentration) and the bottom-$k$ locations. On one hand, these estimates provide more information about the blackbox function than just the maximizers or minimizers in BO. On the other hand, they may require less resource (i.e., evaluations of the blackbox function) than estimating the entire function in ED. Besides, as the top-$k$ locations consist of exactly $k$ locations in the field, it circumvents the issue of undesirably large or small level sets in LSE.

Our main contribution in this paper is to formulate the above challenge and resolve it with a theoretically grounded solution. Specifically, we propose *the active set ordering problem* to capture the above scenario (Sec. 2.1). It aims to estimate subsets of the input domain that are defined based on pairwise comparisons/orderings between the blackbox function evaluations.[1] These subsets include the maximizers, the minimizers, and the top-$k$ inputs with the highest function evaluations. Like Bayesian ED, BO, and LSE, we adopt the pool-based active learning setting [18] in constructing a solution that sequentially selects a *sampling input* from the domain at each iteration. The knowledge from observing function evaluations at the sampling inputs helps predicting the subsets of interest and directs the algorithm to select the next sampling input. To facilitate the presentation of our method, we begin with the building block of our ordering-based problem: pairwise comparison/ordering between function evaluations in Sec. 3. Specifically, we propose a new kind of regret to quantify the loss of a pairwise ordering (Sec. 3.1), a prediction of the top-$k$ inputs based on only the posterior mean (Sec. 3.2), and a sampling strategy that is equipped with a theoretical performance guarantee for the proposed prediction (Sec. 3.3). Subsequently, these concepts of the regret, the prediction, and the sampling strategy are extended to orderings between sets, which ultimately addresses the active set ordering problem in Sec. 4. Notably, the regret simplifies to the well-known regret in BO (Remark 4.1). Hence, we recover both the theoretical analysis and the GP-UCB algorithm [19] when considering a special case of our problem setting (Remark 4.5). In Sec. 5, we empirically validate the performance of our solution using several synthetic functions and real-world datasets.

## 2 Preliminaries and Problem Statement

### 2.1 Top-$k$ set

Adopting an assumption in existing *level set estimation* (LSE) works [1, 8], we consider a blackbox function $f : \mathcal{X} \to \mathbb{R}$ where the domain $\mathcal{X}$ is a finite set of $n$ elements in $\mathbb{R}^d$. Let $\mathcal{S}^c \triangleq \mathcal{X} \setminus \mathcal{S}$ denote *the complement* of any subset $\mathcal{S} \subset \mathcal{X}$. In this paper, *the ordering between inputs* are determined with respect to their corresponding blackbox function evaluations. Hence, we use the term "the ordering between $\mathbf{x}$ and $\mathbf{x}'$" and "the ordering between $f(\mathbf{x})$ and $f(\mathbf{x}')$" interchangeably.

**Definition 2.1** (Top-$k$ set). The *top-$k$ set*, denoted as $\mathcal{S}(k)$, is the set of $k$ inputs with the highest function evaluations. Specifically, $|\mathcal{S}(k)| = k$ and $\forall \mathbf{x} \in \mathcal{S}(k)$, $\forall \mathbf{x}' \in \mathcal{S}^c(k)$, $f(\mathbf{x}) \geq f(\mathbf{x}')$.

In this work, we propose *the active set ordering problem* to estimate the top-$k$ set $\mathcal{S}(k)$ of a blackbox function $f$ by efficiently gathering *noisy function evaluations* in a sequential manner. Furthermore, it includes the *Bayesian optimization* (BO) problem when $k = 1$ because the top-1 set $\mathcal{S}(1)$ contains a maximizer of $f$.

### 2.2 Gaussian Process

The noisy function evaluation mentioned in the previous section is denoted as $y(\mathbf{x}) \triangleq f(\mathbf{x}) + \epsilon(\mathbf{x})$ where the noise $\epsilon(\mathbf{x}) \sim \mathcal{N}(0, \sigma_n^2)$ is a Gaussian random variable with a known (or estimated) variance $\sigma_n^2$. To obtain the posterior distribution of the unknown function $f$ given these noisy evaluations, we model $f$ using a *Gaussian process* (GP), that is, every subset of $\{f(\mathbf{x})\}_{\mathbf{x} \in \mathcal{X}}$ follows a multivariate Gaussian distribution [16]. A GP is fully specified by its prior mean and its kernel $k_{\mathbf{x}, \mathbf{x}'} \triangleq \mathrm{cov}(f(\mathbf{x}), f(\mathbf{x}'))$ which measures the covariance between function values. Let $\mathcal{D}_t$ denote the set of sampling inputs in the first $t - 1$ iterations. Then, given $\mathbf{y}_{\mathcal{D}_t} \triangleq (y(\mathbf{x}))_{\mathbf{x} \in \mathcal{D}_t}$, the predictive distribution of any function evaluation $f(\mathbf{x})$ follows a Gaussian distribution with the following mean and variance:

$$\mu_t(\mathbf{x}) \triangleq \mathbf{k}_t(\mathbf{x})^\top (\mathbf{K}_t + \sigma_n^2 \mathbf{I})^{-1} \mathbf{y}_{\mathcal{D}_t} \qquad \sigma_t^2(\mathbf{x}) \triangleq k(\mathbf{x}, \mathbf{x}) - \mathbf{k}_t(\mathbf{x})^\top (\mathbf{K}_t + \sigma_n^2 \mathbf{I})^{-1} \mathbf{k}_t(\mathbf{x})$$

where $\mathbf{k}_t(\mathbf{x}) \triangleq (k(\mathbf{x}, \mathbf{x}'))_{\mathbf{x}' \in \mathcal{D}_t}$, $\mathbf{K}_t \triangleq (k(\mathbf{x}, \mathbf{x}'))_{\mathbf{x}, \mathbf{x}' \in \mathcal{D}_t}$, and $\mathbf{I}$ is the identity matrix [16].

Assuming $f$ belongs to a *reproducing kernel Hilbert space* with its norm bounded by $B > 0$, due to [5], we have the following confidence bound of $f(\mathbf{x})$.[2]

---

[1]Existing research on best-$k$ arm identification in the multi-armed bandit literature poses a similar problem, but it often focuses on the pure exploration setting and assumes independent arms which are inadequate for modelling blackbox functions [10–12].

[2]Alternatively, we may consider using the confidence bounds established in [19].

**Lemma 2.2.** *Pick $\delta \in (0,1)$ and set $\beta_t = (B + \sigma_n \sqrt{2(\gamma_{t-1} + 1 + \log 1/\delta)})^2$. Then, the following event happens with probability of at least $1 - \delta$,*

$$\forall \mathbf{x} \in \mathcal{X}, \ \forall t \geq 1, \ l_t(\mathbf{x}) \leq f(\mathbf{x}) \leq u_t(\mathbf{x})$$

*where $l_t(\mathbf{x}) \triangleq \mu_t(\mathbf{x}) - \beta_t^{1/2} \sigma_t(\mathbf{x})$, $u_t(\mathbf{x}) \triangleq \mu_t(\mathbf{x}) + \beta_t^{1/2} \sigma_t(\mathbf{x})$, and $\gamma_{t-1} \triangleq \max_{A \subset \mathcal{X}: |A| = t-1} I(\mathbf{y}_A; \mathbf{f}_A)$ is the maximum information gain of $\mathbf{f}_A \triangleq \{f(\mathbf{x})\}_{\mathbf{x} \in A}$ through observing $\mathbf{y}_A$ over all subsets $A \subset \mathcal{X}$ of size $|A| = t - 1$.*

To ease notational clutter, we denote the above confidence interval of $f(\mathbf{x})$ as $\mathcal{C}_t(\mathbf{x}) \triangleq [l_t(\mathbf{x}), u_t(\mathbf{x})]$ and its length as $|\mathcal{C}_t(\mathbf{x})| \triangleq u_t(\mathbf{x}) - l_t(\mathbf{x})$.

# 3 Active Pairwise Ordering: $n = 2$

From Definition 2.1, *pairwise orderings* (or pairwise comparisons) are the building blocks of our active set ordering problem. Hence, to facilitate the exposition of the key ideas, let us begin with a simplistic setting where the input domain $\mathcal{X}$ consists of only $n = 2$ inputs, i.e., $\mathcal{X} = \{\tilde{\mathbf{x}}, \tilde{\mathbf{x}}'\}$ and $f(\tilde{\mathbf{x}}) \neq f(\tilde{\mathbf{x}}')$. The problem is to determine the top-1 set $\mathcal{S}(1)$, i.e., the maximizer of $f$ (equivalently, the minimizer of $f$). In essence, the goal is to check if $f(\tilde{\mathbf{x}}) > f(\tilde{\mathbf{x}}')$ by strategically collecting noisy evaluations $y(\tilde{\mathbf{x}})$ and $y(\tilde{\mathbf{x}}')$. In particular, at iteration $t$, the algorithm proposes a *sampling input* $\mathbf{x}_t \in \mathcal{X}$ to obtain a noisy evaluation $y(\mathbf{x}_t)$. Then, the GP posterior distribution of $f$ is updated and used to construct a predicted ordering between $\tilde{\mathbf{x}}$ and $\tilde{\mathbf{x}}'$ (i.e., the ordering between $f(\tilde{\mathbf{x}})$ and $f(\tilde{\mathbf{x}}')$). The problem boils down to the strategy of selecting the sampling input $\mathbf{x}_t$ such that *a performance metric* of the predicted ordering between $f(\tilde{\mathbf{x}})$ and $f(\tilde{\mathbf{x}}')$ is satisfactory. In the next section, we introduce a regret definition to serve as a performance metric.

## 3.1 Regret

Let us denote the (unknown) *true ordering* between $\tilde{\mathbf{x}}$ and $\tilde{\mathbf{x}}'$ according to the evaluations of the blackbox function $f$ as $\pi_*$:

$$\pi_*(\tilde{\mathbf{x}}, \tilde{\mathbf{x}}') \triangleq \mathbb{1}_{f(\tilde{\mathbf{x}}) \geq f(\tilde{\mathbf{x}}')} \tag{1}$$

where the indicator function $\mathbb{1}_{f(\tilde{\mathbf{x}}) \geq f(\tilde{\mathbf{x}}')} = 1$ if $f(\tilde{\mathbf{x}}) \geq f(\tilde{\mathbf{x}}')$ and $0$ otherwise. For any ordering $\pi : \{(\tilde{\mathbf{x}}, \tilde{\mathbf{x}}')\} \to \{0, 1\}$, we define the following regret of $\pi(\tilde{\mathbf{x}}, \tilde{\mathbf{x}}')$:

$$r_{\pi(\tilde{\mathbf{x}}, \tilde{\mathbf{x}}')} \triangleq \max\left(0, (2\pi(\tilde{\mathbf{x}}, \tilde{\mathbf{x}}') - 1)(f(\tilde{\mathbf{x}}') - f(\tilde{\mathbf{x}}))\right). \tag{2}$$

In particular, $r_{\pi(\tilde{\mathbf{x}}, \tilde{\mathbf{x}}')=1} = \max(0, f(\tilde{\mathbf{x}}') - f(\tilde{\mathbf{x}}))$ and $r_{\pi(\tilde{\mathbf{x}}, \tilde{\mathbf{x}}')=0} = \max(0, f(\tilde{\mathbf{x}}) - f(\tilde{\mathbf{x}}'))$. The rationale is to ensure poor performance leads to large regret: The regret is $|f(\tilde{\mathbf{x}}) - f(\tilde{\mathbf{x}}')|$ (which increases as the gap between $f(\tilde{\mathbf{x}})$ and $f(\tilde{\mathbf{x}}')$ increases) if the ordering $\pi$ does not align with the true ordering, i.e., $\pi(\tilde{\mathbf{x}}, \tilde{\mathbf{x}}') \neq \pi_*(\tilde{\mathbf{x}}, \tilde{\mathbf{x}}')$. On the contrary, the regret is $0$ (i.e., the best performance) if $\pi(\tilde{\mathbf{x}}, \tilde{\mathbf{x}}') = \pi_*(\tilde{\mathbf{x}}, \tilde{\mathbf{x}}')$.

However, the blackbox function $f$ renders the evaluation of $r_{\pi(\tilde{\mathbf{x}}, \tilde{\mathbf{x}}')}$ impossible. Thus, we resort to relying on the GP posterior distribution of $f$ at iteration $t$ to construct an upper bound of the above regret in the following lemma (proof in Appendix A).

**Lemma 3.1.** *For all $t \geq 1$, let us define*

$$\rho_{\pi(\tilde{\mathbf{x}}, \tilde{\mathbf{x}}')}^{(t)} \triangleq \begin{cases} \max(0, u_t(\tilde{\mathbf{x}}') - l_t(\tilde{\mathbf{x}})) & \textit{if } \pi(\tilde{\mathbf{x}}, \tilde{\mathbf{x}}') = 1 \\ \max(0, u_t(\tilde{\mathbf{x}}) - l_t(\tilde{\mathbf{x}}')) & \textit{if } \pi(\tilde{\mathbf{x}}, \tilde{\mathbf{x}}') = 0 \,. \end{cases} \tag{3}$$

*Then, $\rho_{\pi(\tilde{\mathbf{x}}, \tilde{\mathbf{x}}')}^{(t)}$ is an upper confidence bound of the regret $r_{\pi(\tilde{\mathbf{x}}, \tilde{\mathbf{x}}')}$, i.e.,*

$$P\left(\forall t \geq 1, \ r_{\pi(\tilde{\mathbf{x}}, \tilde{\mathbf{x}}')} \leq \rho_{\pi(\tilde{\mathbf{x}}, \tilde{\mathbf{x}}')}^{(t)}\right) \geq 1 - \delta$$

*where $\delta$ is as defined in Lemma 2.2.*

## 3.2 Prediction

**Definition 3.2** (Predicted pairwise ordering $\pi_{\mu_t}$). Given the above upper bound $\rho^{(t)}_{\pi(\tilde{\mathbf{x}},\tilde{\mathbf{x}}')}$ of the regret, we would like to make a prediction $\pi_{\mu_t}$ that minimizes $\rho^{(t)}_{\pi(\tilde{\mathbf{x}},\tilde{\mathbf{x}}')}$. Therefore, $\pi_{\mu_t}$ is defined as follows

$$\pi_{\mu_t}(\tilde{\mathbf{x}},\tilde{\mathbf{x}}') \triangleq \underset{a\in\{0,1\}}{\operatorname{argmin}} \rho^{(t)}_{\pi(\tilde{\mathbf{x}},\tilde{\mathbf{x}}')=a} \;. \tag{4}$$

As a result, the upper confidence bound of the regret in (3) is minimized at $\pi(\tilde{\mathbf{x}},\tilde{\mathbf{x}}') = \pi_{\mu_t}(\tilde{\mathbf{x}},\tilde{\mathbf{x}}')$ and its minimum value is

$$\rho^{(t)}_{\pi_{\mu_t}(\tilde{\mathbf{x}},\tilde{\mathbf{x}}')} = \max\left(0, \min(u_t(\tilde{\mathbf{x}}') - l_t(\tilde{\mathbf{x}}), u_t(\tilde{\mathbf{x}}) - l_t(\tilde{\mathbf{x}}'))\right) \;. \tag{5}$$

We note that the upper confidence bound of the regret can be interpreted as a measure of the approximation quality or the uncertainty reduction as discussed in the following two remarks.

*Remark* 3.3 (Approximation quality). Since $\rho^{(t)}_{\pi_{\mu_t}(\tilde{\mathbf{x}},\tilde{\mathbf{x}}')} \geq r_{\pi_{\mu_t}(\tilde{\mathbf{x}},\tilde{\mathbf{x}}')}$ for all $t \geq 1$ with probability of at least $1 - \delta$, the regret incurred by the ordering $\pi_{\mu_t}$ cannot exceed *the worst-case regret* $\rho^{(t)}_{\pi_{\mu_t}(\tilde{\mathbf{x}},\tilde{\mathbf{x}}')}$ as shown in Fig. 1a. Hence, if $|f(\tilde{\mathbf{x}}) - f(\tilde{\mathbf{x}}')| > \rho^{(t)}_{\pi_{\mu_t}(\tilde{\mathbf{x}},\tilde{\mathbf{x}}')}$, $\pi_{\mu_t}(\tilde{\mathbf{x}},\tilde{\mathbf{x}}')$ is the true ordering, i.e., $\pi_{\mu_t}(\tilde{\mathbf{x}},\tilde{\mathbf{x}}') = \pi_*(\tilde{\mathbf{x}},\tilde{\mathbf{x}}')$, with probability of at least $1 - \delta$.

*Remark* 3.4 (Minimum uncertainty reduction). In Fig. 1b, one can interpret $\rho^{(t)}_{\pi_{\mu_t}(\tilde{\mathbf{x}},\tilde{\mathbf{x}}')}$ as the minimum amount that the confidence intervals $\mathcal{C}_t(\tilde{\mathbf{x}})$ and $\mathcal{C}_t(\tilde{\mathbf{x}}')$ (representing the uncertainty) reduce so that $r_{\pi_{\mu_t}(\tilde{\mathbf{x}},\tilde{\mathbf{x}}')} = 0$ with probability of at least $1 - \delta$.

Moreover, the prediction $\pi_{\mu_t}$ can be obtained using only the GP posterior mean (proof in Appendix B), which explains the name of our approach: *mean prediction* (MP) and the notation $\pi_{\mu_t}$.

**Lemma 3.5** (Mean prediction). *The predicted ordering $\pi_{\mu_t}$ defined in* (4) *can be determined from the GP posterior mean*

$$\pi_{\mu_t}(\tilde{\mathbf{x}},\tilde{\mathbf{x}}') = \mathbb{1}_{\mu_t(\tilde{\mathbf{x}})\geq\mu_t(\tilde{\mathbf{x}}')} \;. \tag{6}$$

(a) $\rho^{(t)}_{\pi_{\mu_t}(\tilde{\mathbf{x}},\tilde{\mathbf{x}}')}$ is the worst-case regret happening when $f(\tilde{\mathbf{x}}) = l_t(\tilde{\mathbf{x}})$ and $f(\tilde{\mathbf{x}}') = u_t(\tilde{\mathbf{x}}')$.

(b) $r_{\pi_{\mu_t}(\tilde{\mathbf{x}},\tilde{\mathbf{x}}')} = 0$ when $(u_t, l_t)$ are refined to $(u'_t, l'_t)$ following an observation, i.e., the reduction in the uncertainty represented as the sum of the two red dashed segments is at least $\rho^{(t)}_{\pi_{\mu_t}(\tilde{\mathbf{x}},\tilde{\mathbf{x}}')}$.

Figure 1: Interpretations of the upper bound $\rho^{(t)}_{\pi_{\mu_t}(\tilde{\mathbf{x}},\tilde{\mathbf{x}}')}$ when $\pi_{\mu_t}(\tilde{\mathbf{x}},\tilde{\mathbf{x}}') = 1$.

## 3.3 Sampling Strategy

Given the regret in Sec. 3.1 and the predicted ordering $\pi_{\mu_t}(\tilde{\mathbf{x}},\tilde{\mathbf{x}}')$ in Sec. 3.2, we would like to select a *sampling input* $\mathbf{x}_t \in \mathcal{X} = \{\tilde{\mathbf{x}},\tilde{\mathbf{x}}'\}$ such that the regret $r_{\pi_{\mu_t}(\tilde{\mathbf{x}},\tilde{\mathbf{x}}')}$ of the predicted ordering $\pi_{\mu_t}(\tilde{\mathbf{x}},\tilde{\mathbf{x}}')$ reduces quickly.

While $r_{\pi_{\mu_t}(\tilde{\mathbf{x}},\tilde{\mathbf{x}}')}$ is unknown, it is bounded by $\rho^{(t)}_{\pi_{\mu_t}(\tilde{\mathbf{x}},\tilde{\mathbf{x}}')}$ with probability of at least $1 - \delta$. Hence, to reduce $r_{\pi_{\mu_t}(\tilde{\mathbf{x}},\tilde{\mathbf{x}}')}$, we aim to reduce $\rho^{(t)}_{\pi_{\mu_t}(\tilde{\mathbf{x}},\tilde{\mathbf{x}}')}$. It is also noted that observing $y(\mathbf{x}_t)$ decreases the confidence interval $|\mathcal{C}_t(\mathbf{x}_t)|$. Hence, to induce the reduction in the regret $r_{\pi_{\mu_t}(\tilde{\mathbf{x}},\tilde{\mathbf{x}}')}$ through observing $y(\mathbf{x}_t)$, we select $\mathbf{x}_t$ such that its confidence interval $|\mathcal{C}_t(\mathbf{x}_t)| \geq \rho^{(t)}_{\pi_{\mu_t}(\tilde{\mathbf{x}},\tilde{\mathbf{x}}')}$ (which guarantees that $|\mathcal{C}_t(\mathbf{x}_t)| \geq r_{\pi_{\mu_t}(\tilde{\mathbf{x}},\tilde{\mathbf{x}}')}$ with probability of at least $1 - \delta$). For instance, choosing $\mathbf{x}_t = \tilde{\mathbf{x}}$ in the left plot of Fig. 1a satisfies this condition, but it does not in the right plot of Fig. 1a. In the following lemma, we show that the following $4$ choices of the sampling input satisfy the proposed condition (see Appendix C).

**Lemma 3.6.** *Let* $\mathcal{Q}_t \triangleq \{\tilde{\mathbf{x}} \triangledown \tilde{\mathbf{x}}', \tilde{\mathbf{x}} \triangle \tilde{\mathbf{x}}', \tilde{\mathbf{x}} \vee \tilde{\mathbf{x}}', \tilde{\mathbf{x}} \wedge \tilde{\mathbf{x}}'\}$ *denote a set[3] of inputs at iteration* $t$ *where*

$$\tilde{\mathbf{x}} \triangledown \tilde{\mathbf{x}}' \triangleq \underset{\mathbf{x} \in \{\tilde{\mathbf{x}},\tilde{\mathbf{x}}'\}}{\operatorname{argmax}} u_t(\mathbf{x}) \qquad\qquad \tilde{\mathbf{x}} \triangle \tilde{\mathbf{x}}' \triangleq \underset{\mathbf{x} \in \{\tilde{\mathbf{x}},\tilde{\mathbf{x}}'\}}{\operatorname{argmin}} l_t(\mathbf{x})$$

$$\tilde{\mathbf{x}} \vee \tilde{\mathbf{x}}' \triangleq \underset{\mathbf{x} \in \{\tilde{\mathbf{x}},\tilde{\mathbf{x}}'\}}{\operatorname{argmax}} |\mathcal{C}_t(\mathbf{x})| \qquad\qquad \tilde{\mathbf{x}} \wedge \tilde{\mathbf{x}}' \triangleq \underset{\mathbf{x} \in \{\tilde{\mathbf{x}}\triangledown\tilde{\mathbf{x}}',\tilde{\mathbf{x}}\triangle\tilde{\mathbf{x}}'\}}{\operatorname{argmin}} |\mathcal{C}_t(\mathbf{x})| \,.$$

*For any* $\mathbf{x}_t \in \mathcal{Q}_t$, $|\mathcal{C}_t(\mathbf{x}_t)| \geq \rho^{(t)}_{\pi_{\mu_t}(\tilde{\mathbf{x}},\tilde{\mathbf{x}}')}$.

**Theorem 3.7.** *By sampling the input* $\mathbf{x}_t$ *following Lemma 3.6, we obtain the following regret bound*

$$P\left(\forall T \geq 1, \ \forall(\mathbf{x}_t)_{t=1}^T \in \prod_{t=1}^T \mathcal{Q}_t, \ R_T \triangleq \sum_{t=1}^T r_{\pi_{\mu_t}(\tilde{\mathbf{x}},\tilde{\mathbf{x}}')} \leq \mathcal{O}(\sqrt{T\beta_T\gamma_T})\right) \geq 1 - \delta$$

*where* $\beta_T$, $\gamma_T$, *and* $\delta$ *are as defined in Lemma 2.2.*

*Remark* 3.8 (Sublinear cumulative regret). If $\gamma_T$ is sublinear, our average cumulative regret is sublinear. This requirement is similar to most BO and LSE algorithms. It is noted that $\gamma_T$ is sublinear for many popular kernels. For instance, $\gamma_T = \mathcal{O}((\log T)^{d+1})$ for the *squared exponential* (SE) kernel as discussed in [19]. In this case, our cumulative regret bound $R_T \leq \mathcal{O}^*(\sqrt{T(\log T)^{2d}})$ is the same as that of GP-UCB [19] (where $\mathcal{O}^*(\cdot)$ denotes asymptotic expressions up to dimension-independent logarithmic factors and is the dimension of the input).

We defer the pseudocode to the next section when $n \geq 2$.

## 4 Active Set Ordering: $n \geq 2$

In this section, we utilize the results in Sec. 3 to present the *mean prediction* (MP) algorithm for the active set ordering problem with $n \geq 2$.

### 4.1 Regret

When $\mathcal{X}$ consists of $n > 2$ inputs, there are multiple pairwise orderings between inputs in $\mathcal{X}$. We overload the ordering notation $\pi_*$ of the true pairwise ordering between $2$ inputs in (1) to the ordering between 2 sets as follows: for any subsets $\mathcal{X}_0 \subset \mathcal{X}$ and $\mathcal{X}_1 \subset \mathcal{X}_0^c$,

$$\pi_*(\mathcal{X}_0, \mathcal{X}_1) = \begin{cases} 1 & \text{if } \forall \mathbf{x} \in \mathcal{X}_0, \ \forall \mathbf{x}' \in \mathcal{X}_1, \ \pi_*(\mathbf{x}, \mathbf{x}') = 1 \\ 0 & \text{if } \forall \mathbf{x} \in \mathcal{X}_0, \ \forall \mathbf{x}' \in \mathcal{X}_1, \ \pi_*(\mathbf{x}, \mathbf{x}') = 0 \,. \end{cases} \tag{7}$$

It is noted that $\pi_*(\mathcal{X}_0, \mathcal{X}_1)$ remains undefined if the two cases above are not satisfied. However, this situation does not arise in our solution. We define the regret of a set ordering (i.e., multiple pairwise orderings) as the maximum regret of all pairwise orderings:

$$r_{\pi(\mathcal{X}_0,\mathcal{X}_1)=i} \triangleq \max_{(\mathbf{x},\mathbf{x}') \in \mathcal{X}_0 \times \mathcal{X}_1} r_{\pi(\mathbf{x},\mathbf{x}')=i} \quad \forall i \in \{0, 1\}$$

where $r_{\pi(\mathbf{x},\mathbf{x}')}$ is defined in (2) and $\mathcal{X}_0 \times \mathcal{X}_1$ is the Cartesian product of $\mathcal{X}_0$ and $\mathcal{X}_1$, i.e.,

$$r_{\pi(\mathcal{X}_0,\mathcal{X}_1)=i} = \max(0, (2i - 1) \max_{(\mathbf{x},\mathbf{x}') \in \mathcal{X}_0 \times \mathcal{X}_1} (f(\mathbf{x}') - f(\mathbf{x}))) \quad \forall i \in \{0, 1\}. \tag{8}$$

---

[3] For convenience, we allow duplicate elements in $\mathcal{Q}_t$.

*Remark* 4.1. It is noted that $r_{\pi(\mathcal{X}_0,\mathcal{X}_1)}$ coincides with the well-known regret in BO when we consider the problem of predicting a maximizer of $f$. In particularly, predicting $\hat{\mathbf{x}}_*$ as a maximizer of $f$ is equivalent to predicting the set ordering $\pi(\{\hat{\mathbf{x}}_*\}, \mathcal{X} \setminus \{\hat{\mathbf{x}}_*\}) = 1$. Its regret is $r_{\pi(\{\hat{\mathbf{x}}_*\}, \mathcal{X}\setminus\{\hat{\mathbf{x}}_*\})=1} = \max_{\mathbf{x}\in\mathcal{X}} f(\mathbf{x}) - f(\hat{\mathbf{x}}_*)$ as shown in Appendix E.1.

Following the upper confidence bound of the regret of pairwise orderings in (3), we show in Appendix F that with probability of at least $1 - \delta$, for all $t \geq 1$ and for all subsets $\mathcal{X}_0 \subset \mathcal{X}$, $\mathcal{X}_1 \subset \mathcal{X}_0^c$, $r_{\pi(\mathcal{X}_0,\mathcal{X}_1)} \leq \rho_{\pi(\mathcal{X}_0,\mathcal{X}_1)}^{(t)}$ where

$$\rho_{\pi(\mathcal{X}_0,\mathcal{X}_1)}^{(t)} \triangleq \max_{(\mathbf{x},\mathbf{x}')\in\mathcal{X}_0\times\mathcal{X}_1} \rho_{\pi(\mathbf{x},\mathbf{x}')=\pi(\mathcal{X}_0,\mathcal{X}_1)}^{(t)} .$$

## 4.2 Prediction

In this section, we generalize the prediction in Sec. 3.2 to set orderings. From Lemma 3.5, there is no contradiction in the pairwise orderings $\pi_{\mu_t}(\mathbf{x}, \mathbf{x}')$ (defined in (4)) for all $\{\mathbf{x}, \mathbf{x}'\} \subset \mathcal{X}$. In other words, the transitivity property holds for the binary relation $\pi_{\mu_t}$ as shown in Appendix G.

**Definition 4.2** (Predicted top-$k$ set $\mathcal{S}_{\mu_t}(k)$). Let $\mathcal{S}_{\mu_t}(k)$ be a subset of $\mathcal{X}$ such that

$$|\mathcal{S}_{\mu_t}(k)| = k , \quad \pi_{\mu_t}(\mathcal{S}_{\mu_t}(k), \mathcal{S}_{\mu_t}^c(k)) = 1 \tag{9}$$

where the set ordering $\pi_{\mu_t}(\mathcal{S}_{\mu_t}(k), \mathcal{S}_{\mu_t}^c(k))$ is obtained by substituting $\pi_*$ with the pairwise ordering $\pi_{\mu_t}$ (see Definition 3.2) in (7). From Lemma 3.5, $\mathcal{S}_{\mu_t}(k)$ is basically the set of $k$ inputs with the highest GP posterior mean values.

As $\pi_{\mu_t}(\mathcal{S}_{\mu_t}(k), \mathcal{S}_{\mu_t}^c(k)) = 1$ implies that $\pi_{\mu_t}(\mathbf{x}, \mathbf{x}') = 1$ for all $(\mathbf{x}, \mathbf{x}') \in \mathcal{S}_{\mu_t}(k) \times \mathcal{S}_{\mu_t}^c(k)$, the upper confidence bound of the regret is

$$\rho_{\pi_{\mu_t}(\mathcal{S}_{\mu_t}(k),\mathcal{S}_{\mu_t}^c(k))}^{(t)} = \max_{(\mathbf{x},\mathbf{x}')\in\mathcal{S}_{\mu_t}(k)\times\mathcal{S}_{\mu_t}^c(k)} \rho_{\pi_{\mu_t}(\mathbf{x},\mathbf{x}')}^{(t)} . \tag{10}$$

## 4.3 Sampling Strategy

Like in Sec. 3.3, our key idea is to select $\mathbf{x}_t$ such that the length $|\mathcal{C}_t(\mathbf{x}_t)|$ of its confidence interval bounds $\rho_{\pi_{\mu_t}(\mathcal{S}_{\mu_t}(k),\mathcal{S}_{\mu_t}^c(k))}^{(t)}$ (in (10)). Since $\rho_{\pi_{\mu_t}(\mathcal{S}_{\mu_t}(k),\mathcal{S}_{\mu_t}^c(k))}^{(t)}$ is the maximum upper confidence bound of the regret of all pairwise orderings involved in defining $\mathcal{S}_{\mu_t}(k)$, we first determine the input pair $(\bar{\mathbf{x}}_t, \bar{\mathbf{x}}_t')$ that incurs the maximum upper confidence bound of the regret. This is also the input pair that $\pi_{\mu_t}$ most likely makes a mistake, following the intuition from [12].

$$(\bar{\mathbf{x}}_t, \bar{\mathbf{x}}_t') \triangleq \underset{(\mathbf{x},\mathbf{x}')\in\mathcal{S}_{\mu_t}(k)\times\mathcal{S}_{\mu_t}^c(k)}{\operatorname{argmax}} \rho_{\pi_{\mu_t}(\mathbf{x},\mathbf{x}')}^{(t)} . \tag{11}$$

It is noted that $(\bar{\mathbf{x}}_t, \bar{\mathbf{x}}_t')$ is constructed from an input in the predicted top-$k$ set $\mathcal{S}_{\mu_t}(k)$ and an input in its complement $\mathcal{S}_{\mu_t}^c(k)$. As the estimation of the top-$k$ set improves, we expect these 2 inputs to be at both sides of the boundary of the top-$k$ set: inside the top-$k$ set vs. outside the top-$k$ set.

Then, extended from Lemma 3.6, the following lemma shows that the above desirable property is satisfied by choosing the sampling input $\mathbf{x}_t$ from any inputs in the set $\bar{\mathcal{Q}}_t \triangleq \{\bar{\mathbf{x}}_t \triangledown \bar{\mathbf{x}}_t', \bar{\mathbf{x}}_t \triangle \bar{\mathbf{x}}_t', \bar{\mathbf{x}}_t \vee \bar{\mathbf{x}}_t', \bar{\mathbf{x}}_t \wedge \bar{\mathbf{x}}_t'\}$ (defined in Lemma 3.6).

**Lemma 4.3.** *For any $\mathbf{x}_t \in \bar{\mathcal{Q}}_t$, $|\mathcal{C}_t(\mathbf{x}_t)| \geq \rho_{\pi_{\mu_t}(\mathcal{S}_{\mu_t}(k),\mathcal{S}_{\mu_t}^c(k))}^{(t)}$.*

The proof is shown in Appendix H. As a result, the cumulative regret incurred by choosing $\mathbf{x}_t$ in Lemma 4.3 is bounded in the following theorem (proof in Appendix I).

**Theorem 4.4.** *By sampling $\mathbf{x}_t$ following Lemma 4.3, we obtain the following cumulative regret bound*

$$P\left(\forall T \geq 1, \forall(\mathbf{x}_t)_{t=1}^T \in \prod_{i=1}^T \bar{\mathcal{Q}}_t, R_{T,k} \triangleq \sum_{t=1}^T r_{\pi_{\mu_t}(\mathcal{S}_{\mu_t}(k),\mathcal{S}_{\mu_t}^c(k))} \leq \mathcal{O}(\sqrt{T\beta_T\gamma_T})\right) \geq 1 - \delta$$

*where $\beta_T$, $\gamma_T$, and $\delta$ are as defined in Lemma 2.2.*

---

**Algorithm 1** Mean Prediction (MP) for Active Set Ordering

---

**Require:** $\mathcal{X}, \mathcal{D}_0, k, T$

1: **for** $t = 1$ **to** $T$ **do**
2:     Update GP posterior belief: $\{\mu_t(\mathbf{x})\}_{\mathbf{x} \in \mathcal{X}}, \{\sigma_t(\mathbf{x})\}_{\mathbf{x} \in \mathcal{X}}$.
3:     Construct $\mathcal{S}_{\mu_t}(k)$ as top-$k$ inputs with the highest values of $\mu_t$.          ▷ Prediction
4:     $(\bar{\mathbf{x}}_t, \bar{\mathbf{x}}'_t) = \mathrm{argmax}_{(\mathbf{x}, \mathbf{x}') \in \mathcal{S}_{\mu_t}(k) \times \mathcal{S}^c_{\mu_t}(k)} \rho^{(t)}_{\pi_{\mu_t}(\mathbf{x}, \mathbf{x}')}$
5:     Select $\mathbf{x}_t \in \{\bar{\mathbf{x}}_t \triangledown \bar{\mathbf{x}}'_t, \bar{\mathbf{x}}_t \triangle \bar{\mathbf{x}}'_t, \bar{\mathbf{x}}_t \vee \bar{\mathbf{x}}'_t, \bar{\mathbf{x}}_t \wedge \bar{\mathbf{x}}'_t\}$.      ▷ Sampling input
6:     $\mathbf{y}_t(\mathcal{D}_t) \leftarrow \mathbf{y}(\mathcal{D}_{t-1}) \cup \{y(\mathbf{x}_t)\}$
7: **end for**
8: Update GP posterior belief: $\{\mu_{T+1}(\mathbf{x})\}_{\mathbf{x} \in \mathcal{X}}, \{\sigma_{T+1}(\mathbf{x})\}_{\mathbf{x} \in \mathcal{X}}$.
9: Construct $\mathcal{S}_{\mu_{T+1}}(k)$ as top-$k$ inputs with the highest values of $\mu_{T+1}$.
10: **return** $\mathcal{S}_{\mu_{T+1}}(k)$.

---

We call the algorithm that makes prediction using the GP posterior mean and selects the sampling input $\mathbf{x}_t$ following Lemma 4.3 the *mean prediction* (MP) algorithm. Its pseudocode is shown in Algorithm 1. Theorem 4.4 indicates that MP incurs a sublinear cumulative regret for several commonly used kernels with sublinear $\gamma_T$ [19].

*Remark* 4.5 (Bayesian optimization as an active set ordering problem with $k = 1$). We show in Appendix E.2 that when $k = 1$, $\bar{\mathbf{x}}_t \triangledown \bar{\mathbf{x}}'_t \in \mathrm{argmax}_{\mathbf{x} \in \mathcal{X}} u_t(\mathbf{x})$, which is the sampling input in the GP-UCB algorithm [19]. Additionally, as discussed in Sec. 4.1, the regret $r_{\pi(\mathcal{S}(1), \mathcal{S}^c(1))}$ is the well-known regret in BO. Hence, we recover both the GP-UCB algorithm and its regret bound when $k = 1$ (although we consider the regret of the prediction rather than that of the sampling input). Moreover, this new construction of GP-UCB leads to some subtle insights. Firstly, while the GP posterior mean has been used in computing the inference regret of entropy search methods [9, 21], there has not been any theoretical justification for using the posterior mean. In contrast, the theoretical analysis in our work justifies the use of the maximizer of the GP posterior mean as an estimate of the maximizer of $f$. Secondly, by predicting the maximizer using the GP posterior mean, $\bar{\mathbf{x}} \triangledown \bar{\mathbf{x}}'$ is not the only sampling input that achieves a sublinear cumulative regret. In fact, there are other choices of the sampling input as shown in Lemma 4.3. Similarly, we note that the LCB algorithm to find the minimizer of a blackbox function can be recovered by setting $k = n - 1$ and $\mathbf{x}_t = \bar{\mathbf{x}} \triangle \bar{\mathbf{x}}'$.

*Remark* 4.6 (Lower bound of active set ordering problem). Let the lower bound of the active set ordering problem be the lower bound of the cumulative regret of the worst-case problem instance over all possible values of $k$. Then, it should be at least as large as the lower bound of the special case where $k = 1$, which is the BO problem according to Remark 4.5. Furthermore, BO has known lower bounds for several common kernels, e.g., for the SE kernel, the lower bound of the cumulative regret is $\Omega(\sqrt{T(\log T)^{d/2}})$ [17]. Hence, the lower bound of the active set ordering problem is at least $\Omega(\sqrt{T(\log T)^{d/2}})$. Additionally, similar to Remark 3.8, the cumulative regret of our solution in Theorem 4.4 is bounded by $R_T \leq \mathcal{O}^*(\sqrt{T(\log T)^{2d}})$. Hence, it matches the lower bound up to the replacement of $d/2$ by $2d + O(1)$.

*Remark* 4.7. Updating the GP posterior belief incurs $\mathcal{O}(|\mathcal{D}_t|^3 + n|\mathcal{D}_t|^2)$ (including $\mathcal{O}(|\mathcal{D}_t|^3)$ for training and $\mathcal{O}(n|\mathcal{D}_t|^2)$ for prediction). Given the GP posterior belief, Algorithm 1 involves the following 2 major steps. First, in line 3 of Algorithm 1, it takes $\mathcal{O}(n \log k)$ to find the top-$k$ inputs $\mathcal{S}_{\mu_t}(k)$ by using a max heap of size $k$ and scanning through the GP posterior mean of all $n$ inputs. Second, in line 4 of Algorithm 1, it takes $\mathcal{O}(k(n-k))$ to scan through the elements in $\mathcal{S}_{\mu_t}(k) \times \mathcal{S}^c_{\mu_t}(k)$. Therefore, an iteration of Algorithm 1 takes $\mathcal{O}(|\mathcal{D}_t|^3 + n|\mathcal{D}_t|^2 + n \log k + k(n - k))$.

*Remark* 4.8 (Active multiple set ordering). Let us consider the problem of estimating $m$ top-$k$ sets: $\mathcal{S}(k_1), \mathcal{S}(k_2), \ldots, \mathcal{S}(k_m)$ simultaneously (motivated in Sec. 1). This problem is analogous to finding $k$ contour lines of a blackbox function, where each contour line represents the boundary between $\mathcal{S}(k_i)$ and its complement $\mathcal{S}^c(k_i)$. To solve this problem, we define the following input pair

$$(\bar{\bar{\mathbf{x}}}_t, \bar{\bar{\mathbf{x}}}'_t) \triangleq \underset{(\mathbf{x}, \mathbf{x}') \in \left(\cup_{i=1}^m \mathcal{S}_{\mu_t}(k_i) \times \mathcal{S}^c_{\mu_t}(k_i)\right)}{\mathrm{argmax}} \rho^{(t)}_{\pi_{\mu_t}(\mathbf{x}, \mathbf{x}')} . \tag{12}$$

In other words, we aim to reduce the maximum regret incurred by the predicted pairwise orderings in all $m$ top-$k$ sets. Given $(\bar{\bar{\mathbf{x}}}_t, \bar{\bar{\mathbf{x}}}'_t)$ in (12), MP proceeds by sampling the input $\mathbf{x}_t$ according to Lemma 4.3, i.e., $\mathbf{x}_t \in \{\bar{\bar{\mathbf{x}}}_t \triangledown \bar{\bar{\mathbf{x}}}'_t, \bar{\bar{\mathbf{x}}}_t \triangle \bar{\bar{\mathbf{x}}}'_t, \bar{\bar{\mathbf{x}}}_t \vee \bar{\bar{\mathbf{x}}}'_t, \bar{\bar{\mathbf{x}}}_t \wedge \bar{\bar{\mathbf{x}}}'_t\}$. The approach is elaborated in Appendix J.

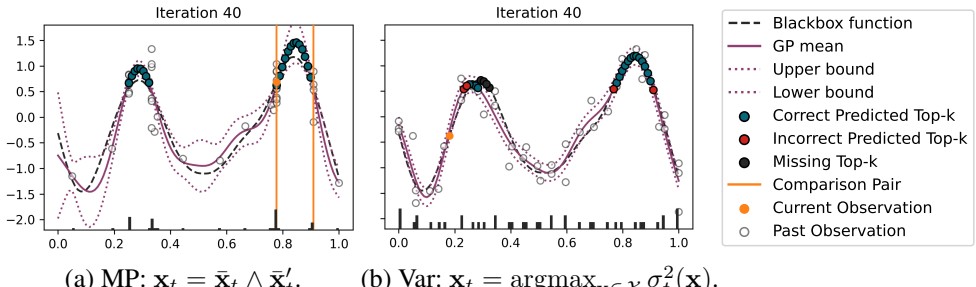

(a) MP: $\mathbf{x}_t = \bar{\mathbf{x}}_t \wedge \bar{\mathbf{x}}_t'$.  (b) Var: $\mathbf{x}_t = \mathrm{argmax}_{\mathbf{x} \in \mathcal{X}} \sigma_t^2(\mathbf{x})$.

Figure 2: Plot of sampling inputs, GP posterior distribution, and the performance of (a) MP and (b) Var in estimating $\mathcal{S}(20)$ of a synthetic function. The comparison pair is $(\bar{\mathbf{x}}_t, \bar{\mathbf{x}}_t')$ in (11). The histogram on the horizontal axis shows the frequency of sampling inputs in 40 iterations.

We also note that active multiple set ordering is able to find both maximizers $\mathcal{S}(1)$ and minimizers $\mathcal{S}^c(n-1)$ simultaneously, a problem has not been studied in GP-UCB [19].

## 5 Experiments

### 5.1 Active Set Ordering

In this section, we validate the empirical performance of our MP algorithm with different choices of the sampling input in Lemma 4.3: $\bar{\mathbf{x}}_t \bigtriangledown \bar{\mathbf{x}}_t'$, $\bar{\mathbf{x}}_t \bigtriangleup \bar{\mathbf{x}}_t'$, $\bar{\mathbf{x}}_t \vee \bar{\mathbf{x}}_t'$, and $\bar{\mathbf{x}}_t \wedge \bar{\mathbf{x}}_t'$ by comparing with 2 baselines: an uncertainty sampling approach, called *Var*, that selects the sampling input with the highest GP posterior variance, i.e., $\mathbf{x}_t \in \mathrm{argmax}_{\mathbf{x} \in \mathcal{X}} \sigma_t^2(\mathbf{x})$, and a baseline, called *Rand*, that selects the sampling input at random. The regret $r_{\pi_{\mu_t}(\mathcal{S}_{\mu_t}(k), \mathcal{S}_{\mu_t}^c(k))}$ is used to measure the performance of each algorithm, i.e., the prediction of the top-$k$ set consists of the $k$ inputs with the highest GP posterior mean.

To begin with, we visualize sampling inputs and the accuracy of $\mathcal{S}_{\mu_t}(20)$ that come from our MP algorithm with $\mathbf{x}_t = \bar{\mathbf{x}}_t \wedge \bar{\mathbf{x}}_t'$ and the Var algorithm in Fig. 2. In Fig. 2a, the histogram shows that the sampling inputs are at the boundary of $\mathcal{S}(20)$. This is highly desirable as it is challenging to decide if an input at the boundary belongs to $\mathcal{S}(20)$. Similarly, the input pair in (11) also consists of inputs around this boundary (depicted as vertical orange lines). On the other hand, precisely estimating the function evaluations of inputs far from the boundary, e.g., inputs around $\mathbf{x} = 0.85$ (in $\mathcal{S}(20)$) and inputs around $\mathbf{x} = 0.1$ (not part of $\mathcal{S}(20)$) is unnecessary. We observe that the uncertainty of the GP posterior distribution at these inputs is high in Fig. 2a. Hence, our MP algorithm is able to efficiently concentrate its sampling budget on important inputs at the boundary of the top-$k$ set. Interestingly, this boundary serves as a contour line of the blackbox function, indicating that our solution could potentially be applied to estimate the contour line by specifying the proportion of the input domain where function evaluations exceed this contour. Regarding the Var algorithm (i.e., uncertainty sampling) in Fig. 2b, the histogram shows that sampling inputs are distributed evenly across the input domain. It is because Var aims to reduce the uncertainty of the function evaluation throughout the input domain without considering the current predicted $\mathcal{S}_{\mu_t}(20)$. For example, it is inefficient to select sampling inputs far away from the boundary of $\mathcal{S}(20)$. It is observed that the estimation of function evaluations at the boundary of $\mathcal{S}(20)$ using Var is more uncertain than that using the MP algorithm given the same number of sampling inputs. This results in erroneously predicting certain inputs in $\mathcal{S}(20)$ (depicted as red dots) and overlooking several inputs in $\mathcal{S}(20)$ (depicted as black dots).[4]

We numerically report the performance using the proposed regret $r_{\pi_{\mu_t}(\mathcal{S}_{\mu_t}(k), \mathcal{S}_{\mu_t}^c(k))}$. The experiments are conducted on 4 synthetic functions: a function sampled from a GP, Branin-Hoo function, Goldstein-Price function with a noise of $\sigma_n = 0.1$, and Hartmann-6D function with a nosie of $\sigma_n = 0.01$ [20]. For the first three synthetic functions, the input domain is discretized into a set of 100 points, whereas for the Hartmann-6D function, it is discretized into a set of 1000 points. Motivated by environmental monitoring problems, we generate 3 active set ordering problems that

---

[4]The complete animation is available in the supplementary materials.

estimate the top-5 set using the dataset of $NO_3$ concentration in the Lake Zurich (downloaded from `https://wldb.ilec.or.jp/Lake/EUR-06/datalist`), the dataset of the phosphorus concentration in the Brooms Barn [22], and the dataset of the humidity in the Intel Lab (downloaded from `https://db.csail.mit.edu/labdata/labdata.html`). The environment field is discretized into a set of 100 locations in the experiments with the $NO_3$ and humidity datasets and 400 locations in the experiment with the phosphorus dataset. The experiments are repeated 15 times to account for the randomness in the generation of the observations. Further details are provided in Appendix K. The average and the standard error of the regret are shown in Figs. 3s:a-g. There is not any significant difference in the performance of MP with different sampling inputs in Lemma 4.3. Nevertheless, the MP algorithm with any choice of the sampling input in $\bar{Q}_t$ outperforms the 2 baselines by converging to lower regret.

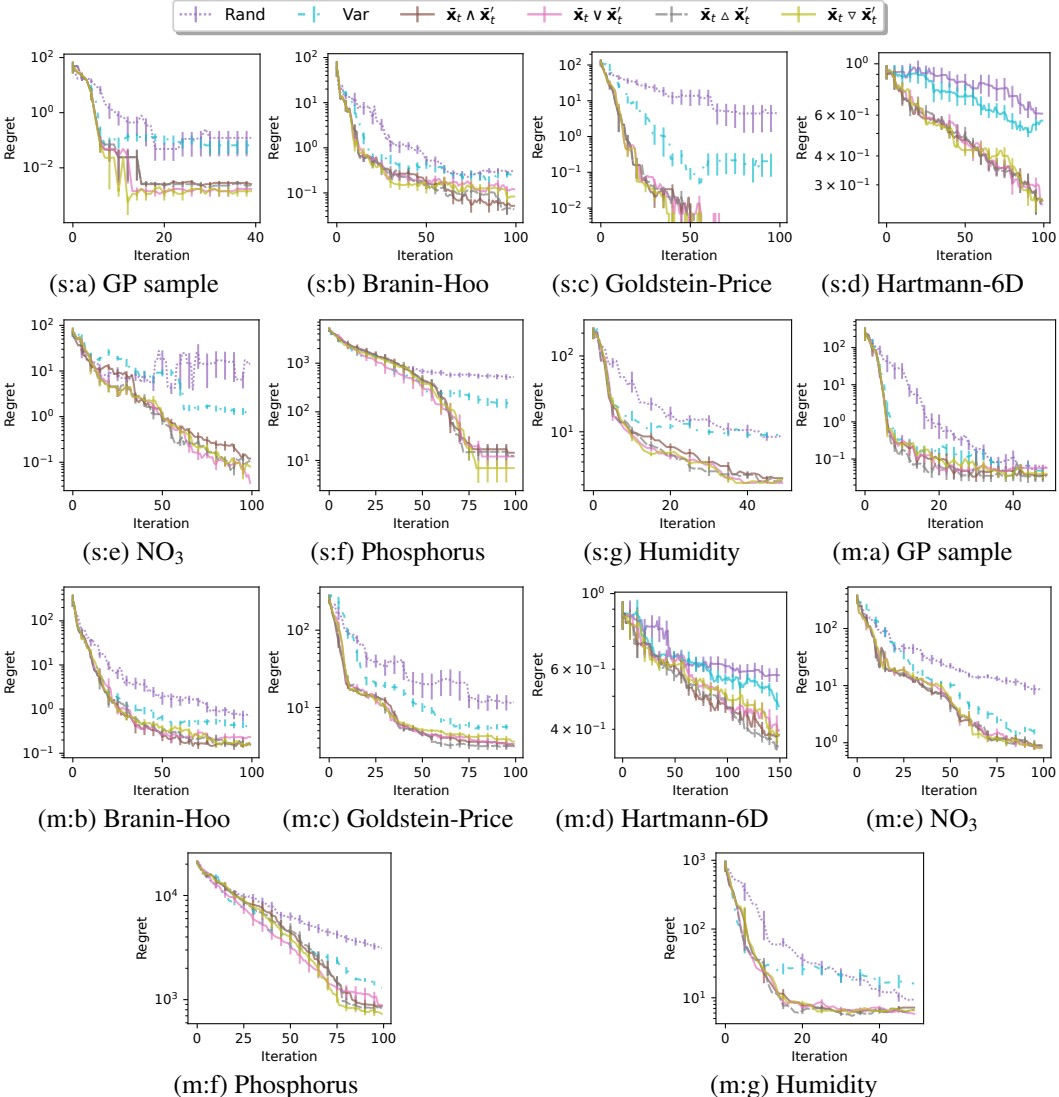

Figure 3: Plots of the regret against the iteration in estimating (s:a-f) the top-5 set $\mathcal{S}(5)$ and (m:a-f) multiple top-$k$ sets: $\mathcal{S}(1)$, $\mathcal{S}(10)$, and $\mathcal{S}(20)$.

## 5.2 Active Multiple Set Ordering

To empirically validate the performance of our MP algorithm in solving the problem of estimating multiple top-$k$ sets, we consider the problem of estimating $\mathcal{S}(1)$ (i.e., maximizers), $\mathcal{S}(10)$, and $\mathcal{S}(20)$ simultaneously, i.e., $k_1 = 1, k_2 = 10, k_3 = 20$ in Remark 4.8. We utilize the same set

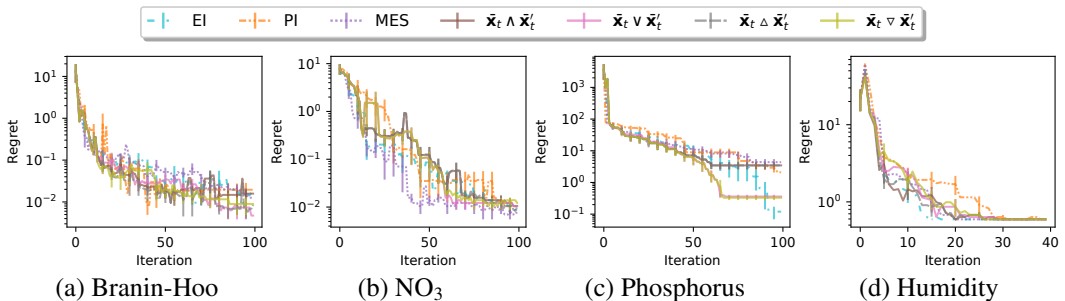

Figure 4: Plots of the regret of the predicted maximizer against the iteration.

of synthetic functions and real-world environmental datasets in the previous section to compare the performance of MP with Var and Rand. The plot of the average and standard error of the maximum regret $\max_{k \in \{1,10,20\}}(r_{\pi_{\mu_t}(\mathcal{S}_{\mu_t}(k),\mathcal{S}^c_{\mu_t}(k))})$ over 15 repeated experiments are shown in Figs. 3m:a-g. The MP algorithm outperforms the other 2 baselines by converging to lower regret. In some active multiple set ordering experiments (e.g., in Figs. 3m:a, 3m:c), the performance gaps between MP and Var are smaller than those in the previous active set ordering experiments (e.g., Figs. 3s:a, 3s:c). It is because estimating multiple top-$k$ sets requires more observations, which makes the performance of MP tend towards that of Var which estimates the entire function.

### 5.3 Bayesian Optimization

When $k = 1$, the active set ordering problem reduces to the BO problem and a sampling input of our MP algorithm, i.e., $\mathbf{x}_t = \bar{\mathbf{x}}_t \triangledown \bar{\mathbf{x}}'_t$, is the same as that of GP-UCB [19]. Therefore, this section empirically demonstrates the performance of MP with different sampling inputs ($\mathbf{x}_t \in \{\bar{\mathbf{x}}_t \triangledown \bar{\mathbf{x}}'_t, \bar{\mathbf{x}}_t \triangle \bar{\mathbf{x}}'_t, \bar{\mathbf{x}}_t \vee \bar{\mathbf{x}}'_t, \bar{\mathbf{x}}_t \wedge \bar{\mathbf{x}}'_t\}$) in solving BO. Our aim is not to show that MP achieves the state-of-the-art performance as a BO solver, but rather to demonstrate that it performs comparably to the well-known GP-UCB algorithm. In addition to comparing with GP-UCB (equivalently, MP with $\mathbf{x}_t = \bar{\mathbf{x}}_t \triangledown \bar{\mathbf{x}}'_t$), we also compare with 3 classical BO solutions: *probability of improvement* (PI) [13], *expected improvement* (EI) [15], and *max-value entropy search* (MES) [21]. The average and the standard error of the regret $r_{\pi_{\mu_t}(\mathcal{S}_{\mu_t}(1),\mathcal{S}^c_{\mu_t}(1))}$ over 15 repeated experiments are shown in Fig. 4. We observe that MP performs comparably with the well-known GP-UCB algorithm (labelled as $\bar{\mathbf{x}}_t \triangledown \bar{\mathbf{x}}'_t$). Expectedly, EI and MES outperform GP-UCB (and hence, MP) in some experiments such as in Figs. 4b and 4c.

## 6 Conclusion

This paper presents a new problem formulation, namely *active set ordering*, that aims to balance between the expensive estimation of the entire function in ED and that of only the maximizers in BO. We propose the *mean prediction* (MP) algorithm to address this problem with a theoretical no-regret guarantee. Interestingly, BO can be framed as a special instance of active set ordering, which leads to several new subtle understandings regarding the predicted maximizer and other alternative sampling inputs. Last, the performance of MP is empirically evaluated using various synthetic functions and real-world datasets.

## Acknowledgments and Disclosure of Funding

This research/project is supported by the National Research Foundation Singapore and DSO National Laboratories under the AI Singapore Programme (AISG Award No: AISG2-RP-2020-018).

DesCartes: this research is supported by the National Research Foundation, Prime Minister's Office, Singapore under its Campus for Research Excellence and Technological Enterprise (CREATE) programme.

This research was partially supported by the Australian Government through the Australian Research Council's Discovery Projects funding scheme (project DP210102798). The views expressed herein are those of the authors and are not necessarily those of the Australian Government or Australian Research Council.

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

## A  Proof of Lemma 3.1

We will show that $\rho_{\pi(\tilde{\mathbf{x}},\tilde{\mathbf{x}}')}^{(t)}$ is an upper confidence bound of the regret $r_{\pi(\tilde{\mathbf{x}},\tilde{\mathbf{x}}')}$, i.e.,

$$P\left(\forall t \geq 1,\ r_{\pi(\tilde{\mathbf{x}},\tilde{\mathbf{x}}')} \leq \rho_{\pi(\tilde{\mathbf{x}},\tilde{\mathbf{x}}')}^{(t)}\right) \geq 1 - \delta\,.$$

The regret $r_{\pi(\tilde{\mathbf{x}},\tilde{\mathbf{x}}')}$ is defined as follows

$$r_{\pi(\tilde{\mathbf{x}},\tilde{\mathbf{x}}')} \triangleq \max\left(0, (2\pi(\tilde{\mathbf{x}},\tilde{\mathbf{x}}') - 1)(f(\tilde{\mathbf{x}}') - f(\tilde{\mathbf{x}}))\right)\,.$$

Furthermore, from Lemma 2.2, with probability of at least $1 - \delta$, for all $t \geq 1$,

$$l_t(\tilde{\mathbf{x}}) \leq f(\tilde{\mathbf{x}}) \leq u_t(\tilde{\mathbf{x}})$$
$$l_t(\tilde{\mathbf{x}}') \leq f(\tilde{\mathbf{x}}') \leq u_t(\tilde{\mathbf{x}}')$$

Hence, with probability of at least $1 - \delta$, for all $t \geq 1$,

$$f(\tilde{\mathbf{x}}) - f(\tilde{\mathbf{x}}') \leq u_t(\tilde{\mathbf{x}}) - l_t(\tilde{\mathbf{x}}')$$
$$f(\tilde{\mathbf{x}}') - f(\tilde{\mathbf{x}}) \leq u_t(\tilde{\mathbf{x}}') - l_t(\tilde{\mathbf{x}})$$

Multiplying both sides with $(2\pi(\tilde{\mathbf{x}},\tilde{\mathbf{x}}') - 1)$, we obtain

$$(2\pi(\tilde{\mathbf{x}},\tilde{\mathbf{x}}') - 1)(f(\tilde{\mathbf{x}}') - f(\tilde{\mathbf{x}})) \leq \begin{cases} u_t(\tilde{\mathbf{x}}) - l_t(\tilde{\mathbf{x}}') & \text{if } \pi(\tilde{\mathbf{x}},\tilde{\mathbf{x}}') = 0 \\ u_t(\tilde{\mathbf{x}}') - l_t(\tilde{\mathbf{x}}) & \text{if } \pi(\tilde{\mathbf{x}},\tilde{\mathbf{x}}') = 1 \end{cases} \triangleq \rho_{\pi(\tilde{\mathbf{x}},\tilde{\mathbf{x}}')}^{(t)}\,.$$

Therefore, with probability of at least $1 - \delta$, for all $t \geq 1$,

$$r_{\pi(\tilde{\mathbf{x}},\tilde{\mathbf{x}}')} \triangleq \max\left(0, (2\pi(\tilde{\mathbf{x}},\tilde{\mathbf{x}}') - 1)(f(\tilde{\mathbf{x}}') - f(\tilde{\mathbf{x}}))\right) \leq \begin{cases} \max(0, u_t(\tilde{\mathbf{x}}) - l_t(\tilde{\mathbf{x}}')) & \text{if } \pi(\tilde{\mathbf{x}},\tilde{\mathbf{x}}') = 0 \\ \max(0, u_t(\tilde{\mathbf{x}}') - l_t(\tilde{\mathbf{x}})) & \text{if } \pi(\tilde{\mathbf{x}},\tilde{\mathbf{x}}') = 1\,. \end{cases}$$

## B  Proof of Lemma 3.5

We note that $\mathbb{1}_{\mu_t(\tilde{\mathbf{x}}) \geq \mu_t(\tilde{\mathbf{x}}')} = 1$ happens when

$$\mu_t(\tilde{\mathbf{x}}) \geq \mu_t(\tilde{\mathbf{x}}')$$
$$\Leftrightarrow \frac{u_t(\tilde{\mathbf{x}}) + l_t(\tilde{\mathbf{x}})}{2} \geq \frac{u_t(\tilde{\mathbf{x}}') + l_t(\tilde{\mathbf{x}}')}{2}$$
$$\Leftrightarrow u_t(\tilde{\mathbf{x}}) - l_t(\tilde{\mathbf{x}}') \geq u_t(\tilde{\mathbf{x}}') - l_t(\tilde{\mathbf{x}})$$
$$\Leftrightarrow \max(0, u_t(\tilde{\mathbf{x}}) - l_t(\tilde{\mathbf{x}}')) \geq \max(0, u_t(\tilde{\mathbf{x}}') - l_t(\tilde{\mathbf{x}}))$$
$$\Leftrightarrow \rho_{\pi(\tilde{\mathbf{x}},\tilde{\mathbf{x}}')=0}^{(t)} \geq \rho_{\pi(\tilde{\mathbf{x}},\tilde{\mathbf{x}}')=1}^{(t)}\,.$$

Therefore,

$$\mathbb{1}_{\mu_t(\tilde{\mathbf{x}}) \geq \mu_t(\tilde{\mathbf{x}}')} = \mathbb{1}_{\rho_{\pi(\tilde{\mathbf{x}},\tilde{\mathbf{x}}')=0}^{(t)} \geq \rho_{\pi(\tilde{\mathbf{x}},\tilde{\mathbf{x}}')=1}^{(t)}} = \underset{a \in \{0,1\}}{\operatorname{argmin}} \rho_{\pi(\tilde{\mathbf{x}},\tilde{\mathbf{x}}')=a}^{(t)} = \pi_{\mu_t}(\tilde{\mathbf{x}}, \tilde{\mathbf{x}}')\,.$$

## C  Proof of Lemma 3.6

We will show that $\rho_{\pi_{\mu_t}(\tilde{\mathbf{x}},\tilde{\mathbf{x}}')}^{(t)} \leq |\mathcal{C}_t(\mathbf{x}_t)|$ when $\mathbf{x}_t$ is taken from the set $\mathcal{Q}_t \triangleq \{\tilde{\mathbf{x}} \triangledown \tilde{\mathbf{x}}', \tilde{\mathbf{x}} \triangle \tilde{\mathbf{x}}', \tilde{\mathbf{x}} \vee \tilde{\mathbf{x}}', \tilde{\mathbf{x}} \wedge \tilde{\mathbf{x}}'\}$.

**Case 1:** $\mathbf{x}_t = \tilde{\mathbf{x}} \triangledown \tilde{\mathbf{x}}' \triangleq \operatorname{argmax}_{\mathbf{x} \in \{\tilde{\mathbf{x}},\tilde{\mathbf{x}}'\}} u_t(\mathbf{x})$

$$\rho_{\pi_{\mu_t}(\tilde{\mathbf{x}},\tilde{\mathbf{x}}')}^{(t)} = \max(0, \min(u_t(\tilde{\mathbf{x}}') - l_t(\tilde{\mathbf{x}}), u_t(\tilde{\mathbf{x}}) - l_t(\tilde{\mathbf{x}}')))$$

$$\leq \max(0, \min(u_t(\tilde{\mathbf{x}} \triangledown \tilde{\mathbf{x}}') - l_t(\tilde{\mathbf{x}}), u_t(\tilde{\mathbf{x}} \triangledown \tilde{\mathbf{x}}') - l_t(\tilde{\mathbf{x}}'))) \tag{13}$$

$$\leq \max(0, u_t(\tilde{\mathbf{x}} \triangledown \tilde{\mathbf{x}}') - l_t(\tilde{\mathbf{x}} \triangledown \tilde{\mathbf{x}}')) \tag{14}$$

$$= u_t(\tilde{\mathbf{x}} \triangledown \tilde{\mathbf{x}}') - l_t(\tilde{\mathbf{x}} \triangledown \tilde{\mathbf{x}}') \tag{15}$$

$$= |\mathcal{C}_t(\tilde{\mathbf{x}} \triangledown \tilde{\mathbf{x}}')| \tag{16}$$

where (13) is because $\tilde{\mathbf{x}} \triangledown \tilde{\mathbf{x}}' \triangleq \mathrm{argmax}_{\mathbf{x}\in\{\tilde{\mathbf{x}},\tilde{\mathbf{x}}'\}} u_t(\mathbf{x})$, (14) is because $\tilde{\mathbf{x}} \triangledown \tilde{\mathbf{x}}' \in \{\tilde{\mathbf{x}}, \tilde{\mathbf{x}}'\}$, (15) is because $u_t(\tilde{\mathbf{x}} \triangledown \tilde{\mathbf{x}}') - l_t(\tilde{\mathbf{x}} \triangledown \tilde{\mathbf{x}}') \geq 0$.

**Case 2:** $\mathbf{x}_t = \tilde{\mathbf{x}} \triangle \tilde{\mathbf{x}}' \triangleq \mathrm{argmin}_{\mathbf{x}\in\{\tilde{\mathbf{x}},\tilde{\mathbf{x}}'\}} l_t(\mathbf{x})$

$$\rho^{(t)}_{\pi_{\mu_t}(\tilde{\mathbf{x}},\tilde{\mathbf{x}}')} = \max(0, \min(u_t(\tilde{\mathbf{x}}') - l_t(\tilde{\mathbf{x}}), u_t(\tilde{\mathbf{x}}) - l_t(\tilde{\mathbf{x}}'))) \tag{17}$$

$$\leq \max(0, \min(u_t(\tilde{\mathbf{x}}') - l_t(\tilde{\mathbf{x}} \triangle \tilde{\mathbf{x}}'), u_t(\tilde{\mathbf{x}}) - l_t(\tilde{\mathbf{x}} \triangle \tilde{\mathbf{x}}'))) \tag{18}$$

$$\leq \max(0, u_t(\tilde{\mathbf{x}} \triangle \tilde{\mathbf{x}}') - l_t(\tilde{\mathbf{x}} \triangle \tilde{\mathbf{x}}')) \tag{19}$$

$$= u_t(\tilde{\mathbf{x}} \triangle \tilde{\mathbf{x}}') - l_t(\tilde{\mathbf{x}} \triangle \tilde{\mathbf{x}}') \tag{20}$$

$$= |\mathcal{C}_t(\tilde{\mathbf{x}} \triangle \tilde{\mathbf{x}}')| \tag{21}$$

where (18) is because $\tilde{\mathbf{x}} \triangle \tilde{\mathbf{x}}' \triangleq \mathrm{argmin}_{\mathbf{x}\in\{\tilde{\mathbf{x}},\tilde{\mathbf{x}}'\}} l_t(\mathbf{x})$, (19) is because $\tilde{\mathbf{x}} \triangle \tilde{\mathbf{x}}' \in \{\tilde{\mathbf{x}}, \tilde{\mathbf{x}}'\}$, (20) is because $u_t(\tilde{\mathbf{x}} \triangle \tilde{\mathbf{x}}') - l_t(\tilde{\mathbf{x}} \triangle \tilde{\mathbf{x}}') \geq 0$.

**Case 3:** $\mathbf{x}_t = \tilde{\mathbf{x}} \vee \tilde{\mathbf{x}}' \triangleq \mathrm{argmax}_{\mathbf{x}\in\{\tilde{\mathbf{x}},\tilde{\mathbf{x}}'\}} |\mathcal{C}_t(\mathbf{x})|$, i.e.,

$$|\mathcal{C}_t(\tilde{\mathbf{x}} \vee \tilde{\mathbf{x}}')| = \max_{\mathbf{x}\in\{\tilde{\mathbf{x}},\tilde{\mathbf{x}}'\}} |\mathcal{C}_t(\mathbf{x})| \tag{22}$$

$$\geq |\mathcal{C}_t(\tilde{\mathbf{x}} \triangle \tilde{\mathbf{x}}')| \tag{23}$$

$$\geq \rho^{(t)}_{\pi_{\mu_t}(\tilde{\mathbf{x}},\tilde{\mathbf{x}}')} \tag{24}$$

where (23) is because $\tilde{\mathbf{x}} \triangle \tilde{\mathbf{x}}' \in \{\tilde{\mathbf{x}}, \tilde{\mathbf{x}}'\}$, (24) is from the above proof of case 2 in (21).

**Case 4:** $\mathbf{x}_t = \tilde{\mathbf{x}} \wedge \tilde{\mathbf{x}}' \triangleq \mathrm{argmin}_{\mathbf{x}\in\{\tilde{\mathbf{x}} \triangledown \tilde{\mathbf{x}}', \tilde{\mathbf{x}}\triangle\tilde{\mathbf{x}}'\}} |\mathcal{C}_t(\mathbf{x})|$. From (16) and (21), it follows that $|\mathcal{C}_t(\tilde{\mathbf{x}} \wedge \tilde{\mathbf{x}}')| \geq \rho^{(t)}_{\pi_{\mu_t}(\tilde{\mathbf{x}},\tilde{\mathbf{x}}')}$.

## D   Proof of Theorem 3.7

By choosing $\mathbf{x}_t$ following Lemma 3.6, with probability of at least $1 - \delta$, for all $t \geq 1$,

$$r_{\pi_{\mu_t}(\tilde{\mathbf{x}},\tilde{\mathbf{x}}')} \leq \rho^{(t)}_{\pi_{\mu_t}(\tilde{\mathbf{x}},\tilde{\mathbf{x}}')} \leq |\mathcal{C}_t(\mathbf{x}_t)| = 2\beta_t^{1/2}\sigma_t(\mathbf{x}_t)$$

Hence, with probability of at least $1 - \delta$, for all $T \geq 1$,

$$\sum_{t=1}^{T} r_{\pi_{\mu_t}(\tilde{\mathbf{x}},\tilde{\mathbf{x}}')} \leq \sum_{t=1}^{T} 2\beta_t^{1/2}\sigma_t(\mathbf{x}_t)$$

Since $\beta_t$ is a non-decreasing sequence, $\beta_t \leq \beta_T$ for all $t \leq T$. Therefore, with probability of at least $1 - \delta$, for all $T \geq 1$,

$$\sum_{t=1}^{T} r_{\pi_{\mu_t}(\tilde{\mathbf{x}},\tilde{\mathbf{x}}')} \leq 2\beta_T^{1/2} \sum_{t=1}^{T} \sigma_t(\mathbf{x}_t)$$

From Lemma 4 in [5],

$$\sum_{t=1}^{T} \sigma_t(\mathbf{x}_t) \leq \sqrt{4(T+2)\gamma_T} = \mathcal{O}(\sqrt{T\gamma_T})$$

Hence, with probability of at least $1 - \delta$, for all $T \geq 1$,

$$\sum_{t=1}^{T} r_{\pi_{\mu_t}(\tilde{\mathbf{x}},\tilde{\mathbf{x}}')} \leq \mathcal{O}(\sqrt{T\beta_T\gamma_T}).$$

i.e.,

$$R_T \triangleq \sum_{t=1}^{T} r_{\pi_{\mu_t}(\tilde{\mathbf{x}},\tilde{\mathbf{x}}')} \leq \mathcal{O}(\sqrt{T\beta_T\gamma_T}).$$

# E  Bayesian Optimization as Active Set Ordering with $k = 1$

## E.1  Regret

For a predicted top-1 set $\mathcal{S}_{\mu_t}(1)$, $\pi_{\mu_t}(\mathcal{S}_{\mu_t}(1), \mathcal{S}_{\mu_t}^c(1)) = 1$. Hence, the regret for predicting $\mathcal{S}_{\mu_t}(1)$ is expressed as follows.

$$r_{\pi_{\mu_t}(\mathcal{S}_{\mu_t}(1), \mathcal{S}_{\mu_t}^c(1))} = \max\left( 0, \max_{(\mathbf{x}, \mathbf{x}') \in \mathcal{S}_{\mu_t}(1) \times \mathcal{S}_{\mu_t}^c(1)} f(\mathbf{x}') - f(\mathbf{x}) \right) .$$

Let $\mathcal{S}_{\mu_t}(1) \triangleq \{\hat{\mathbf{x}}_*\}$ and $\mathbf{x}_* \in \operatorname{argmax}_{\mathbf{x} \in \mathcal{X}} f(\mathbf{x})$.

$$r_{\pi_{\mu_t}(\mathcal{S}_{\mu_t}(1), \mathcal{S}_{\mu_t}^c(1))} = \max\left( 0, \max_{\mathbf{x} \in \mathcal{S}_{\mu_t}^c(1)} (f(\mathbf{x}) - f(\hat{\mathbf{x}}_*)) \right)$$

$$= \max\left( 0, \left( \max_{\mathbf{x} \in \mathcal{S}_{\mu_t}^c(1)} f(\mathbf{x}) \right) - f(\hat{\mathbf{x}}_*) \right) .$$

Since $\mathcal{X} = \mathcal{S}_{\mu_t}(1) \cup \mathcal{S}_{\mu_t}^c(1)$, there are 2 cases

- If $\mathbf{x}_* \in \mathcal{S}_{\mu_t}^c(1)$, then $\max_{\mathbf{x} \in \mathcal{S}_{\mu_t}^c(1)} f(\mathbf{x}) = f(\mathbf{x}_*) \geq f(\hat{\mathbf{x}}_*)$ and

$$r_{\pi_{\mu_t}(\mathcal{S}_{\mu_t}(1), \mathcal{S}_{\mu_t}^c(1))} = \max\left( 0, f(\mathbf{x}_*) - f(\hat{\mathbf{x}}_*) \right) = f(\mathbf{x}_*) - f(\hat{\mathbf{x}}_*) .$$

- If $\mathbf{x}_* \in \mathcal{S}_{\mu_t}(1)$, i.e., $\mathbf{x}_* = \hat{\mathbf{x}}_*$, then $\max_{\mathbf{x} \in \mathcal{S}_{\mu_t}^c(1)} f(\mathbf{x}) \leq f(\hat{\mathbf{x}}_*)$ and

$$r_{\pi_{\mu_t}(\mathcal{S}_{\mu_t}(1), \mathcal{S}_{\mu_t}^c(1))} = 0 = f(\mathbf{x}_*) - f(\hat{\mathbf{x}}_*) .$$

## E.2  Sampling Input

We will show that

$$\bar{\mathbf{x}}_t \nabla \bar{\mathbf{x}}_t' \in \operatorname*{argmax}_{\mathbf{x} \in \mathcal{X}} u_t(\mathbf{x}) .$$

When $k = 1$, due to the definition of $\mathcal{S}_{\mu_t}(k)$ in Definition 4.2

$$\bar{\mathbf{x}}_t \in \operatorname*{argmax}_{\mathbf{x} \in \mathcal{X}} \mu_t(\mathbf{x}) .$$

Let

$$\hat{\mathbf{x}}_t \in \operatorname*{argmax}_{\mathbf{x} \in \mathcal{X}} u_t(\mathbf{x}) .$$

We consider the following 2 cases:

**Case 1:** If $u_t(\hat{\mathbf{x}}_t) = u_t(\bar{\mathbf{x}}_t)$, then

$$\bar{\mathbf{x}}_t \nabla \bar{\mathbf{x}}_t' \in \operatorname*{argmax}_{\mathbf{x} \in \mathcal{X}} u_t(\mathbf{x}) .$$

**Case 2:** If $u_t(\hat{\mathbf{x}}_t) \neq u_t(\bar{\mathbf{x}}_t)$ which implies that $\hat{\mathbf{x}}_t \in \mathcal{S}_{\mu_t}^c(k)$ and $u_t(\hat{\mathbf{x}}_t) > u_t(\bar{\mathbf{x}}_t)$, then we prove that $u_t(\hat{\mathbf{x}}_t) = u_t(\bar{\mathbf{x}}_t')$ by contradiction. Assuming that

$$u_t(\hat{\mathbf{x}}_t) > u_t(\bar{\mathbf{x}}_t') . \tag{25}$$

Let us consider the following upper confidence bounds of pairwise orderings:

$$\rho_{\pi_{\mu_t}(\bar{\mathbf{x}}_t, \hat{\mathbf{x}}_t)}^{(t)} = \max(0, u_t(\hat{\mathbf{x}}_t) - l_t(\bar{\mathbf{x}}_t)) = u_t(\hat{\mathbf{x}}_t) - l_t(\bar{\mathbf{x}}_t) > 0$$

$$\rho_{\pi_{\mu_t}(\bar{\mathbf{x}}_t, \bar{\mathbf{x}}_t')}^{(t)} = \max(0, u_t(\bar{\mathbf{x}}_t') - l_t(\bar{\mathbf{x}}_t)) .$$

Furthermore, from the choice of $(\bar{\mathbf{x}}_t, \bar{\mathbf{x}}_t')$ in (11) and $\hat{\mathbf{x}}_t \in \mathcal{S}_{\mu_t}^c(k)$,

$$\rho_{\pi_{\mu_t}(\bar{\mathbf{x}}_t, \bar{\mathbf{x}}_t')}^{(t)} \geq \rho_{\pi_{\mu_t}(\bar{\mathbf{x}}_t, \hat{\mathbf{x}}_t)}^{(t)} .$$

Hence,

$$\max(0, u_t(\bar{\mathbf{x}}_t') - l_t(\bar{\mathbf{x}}_t)) \geq \max(0, u_t(\hat{\mathbf{x}}_t) - l_t(\bar{\mathbf{x}}_t)) = u_t(\hat{\mathbf{x}}_t) - l_t(\bar{\mathbf{x}}_t) > 0$$
$$u_t(\bar{\mathbf{x}}_t') - l_t(\bar{\mathbf{x}}_t) \geq u_t(\hat{\mathbf{x}}_t) - l_t(\bar{\mathbf{x}}_t)$$
$$u_t(\bar{\mathbf{x}}_t') \geq u_t(\hat{\mathbf{x}}_t)$$

which contradicts to the assumption 25. Therefore, $u_t(\hat{\mathbf{x}}_t) = u_t(\bar{\mathbf{x}}_t')$ and

$$\bar{\mathbf{x}}_t \triangledown \bar{\mathbf{x}}_t' \in \underset{\mathbf{x} \in \mathcal{X}}{\arg\max}\, u_t(\mathbf{x}) .$$

# F    Upper Confidence Bound of the Regret $r_{\pi(\mathcal{X}_0, \mathcal{X}_1)}$

$$r_{\pi(\mathcal{X}_0, \mathcal{X}_1)} \triangleq \underset{(\mathbf{x}, \mathbf{x}') \in \mathcal{X}_0 \times \mathcal{X}_1}{\max}\, r_{\pi(\mathbf{x}, \mathbf{x}') = \pi(\mathcal{X}_0, \mathcal{X}_1)}$$

With probability of at least $1 - \delta$, for all $t \geq 1$ and for all $\{\mathbf{x}, \mathbf{x}'\} \subset \mathcal{X}$, $r_{\pi(\mathbf{x}, \mathbf{x}') = \pi(\mathcal{X}_0, \mathcal{X}_1)} \leq \rho_{\pi(\mathbf{x}, \mathbf{x}') = \pi(\mathcal{X}_0, \mathcal{X}_1)}^{(t)}$. Therefore, with probability of at least $1 - \delta$, for all $t \geq 1$ and for all $\{\mathbf{x}, \mathbf{x}'\} \subset \mathcal{X}$,

$$r_{\pi(\mathcal{X}_0, \mathcal{X}_1)} \leq \underset{(\mathbf{x}, \mathbf{x}') \in \mathcal{X}_0 \times \mathcal{X}_1}{\max}\, \rho_{\pi(\mathbf{x}, \mathbf{x}') = \pi(\mathcal{X}_0, \mathcal{X}_1)}^{(t)} \triangleq \rho_{\pi(\mathcal{X}_0, \mathcal{X}_1)}^{(t)} .$$

# G    Transitivity of Pairwise Orderings

The transitivity property of the binary relation $\mu_{\pi_t}$ follows directly from its connection to the GP posterior mean in Lemma 3.5. In particular, we would like to show that if

$$\pi_{\mu_t}(\mathbf{x}, \mathbf{x}') = 1 \quad \pi_{\mu_t}(\mathbf{x}', \mathbf{x}'') = 1 \tag{26}$$

then

$$\pi_{\mu_t}(\mathbf{x}, \mathbf{x}'') = 1 .$$

From Lemma 3.5, the premise (26) implies that

$$\mu_t(\mathbf{x}) \geq \mu_t(\mathbf{x}') \quad \mu_t(\mathbf{x}') \geq \mu_t(\mathbf{x}'')$$

which implies that

$$\mu_t(\mathbf{x}) \geq \mu_t(\mathbf{x}'') .$$

Applying Lemma 3.5 again, we conclude

$$\pi_{\mu_t}(\mathbf{x}, \mathbf{x}'') = 1 .$$

# H    Proof of Lemma 4.3

Applying Lemma 3.6 to the input pair $(\bar{\mathbf{x}}_t, \bar{\mathbf{x}}_t')$, by selecting $\mathbf{x}_t \in \{\bar{\mathbf{x}}_t \triangledown \bar{\mathbf{x}}_t', \bar{\mathbf{x}}_t \triangle \bar{\mathbf{x}}_t', \bar{\mathbf{x}}_t \vee \bar{\mathbf{x}}_t', \bar{\mathbf{x}}_t \wedge \bar{\mathbf{x}}_t'\}$,

$$|\mathcal{C}_t(\mathbf{x}_t)| \geq \rho_{\pi_{\mu_t}(\bar{\mathbf{x}}_t, \bar{\mathbf{x}}_t')}^{(t)} .$$

Furthermore, from the choice of $(\bar{\mathbf{x}}_t, \bar{\mathbf{x}}_t')$ in (11),

$$\rho_{\pi_{\mu_t}(\bar{\mathbf{x}}_t, \bar{\mathbf{x}}_t')}^{(t)} = \underset{(\mathbf{x}, \mathbf{x}') \in \mathcal{S}_{\mu_t}(k) \times \mathcal{S}_{\mu_t}^c(k)}{\max}\, \rho_{\pi_{\mu_t}(\mathbf{x}, \mathbf{x}')}^{(t)} = \rho_{\pi_{\mu_t}(\mathcal{S}_{\mu_t}(k), \mathcal{S}_{\mu_t}^c(k))}^{(t)} .$$

Therefore,

$$|\mathcal{C}_t(\mathbf{x}_t)| \geq \rho_{\pi_{\mu_t}(\mathcal{S}_{\mu_t}(k), \mathcal{S}_{\mu_t}^c(k))}^{(t)} .$$

**Algorithm 2** Mean Prediction (MP) for Active Multiple Set Ordering

---

**Require:** $\mathcal{X}, \mathcal{D}_0, \{k_1, k_2, \ldots, k_m\}, T$

1: **for** $t = 1$ **to** $T$ **do**
2:      Update GP posterior belief: $\{\mu_t(\mathbf{x})\}_{\mathbf{x} \in \mathcal{X}}, \{\sigma_t(\mathbf{x})\}_{\mathbf{x} \in \mathcal{X}}$.
3:      Construct $\{\mathcal{S}_{\mu_t}(k_i)\}_{i=1}^m$ as the collection of $m$ top-$k$ sets predicted using $\mu_t$.    ▷ Prediction
4:      $(\bar{\bar{\mathbf{x}}}_t, \bar{\bar{\mathbf{x}}}_t') \triangleq \arg\max_{(\mathbf{x},\mathbf{x}') \in \cup_{i=1}^m \mathcal{S}_{\mu_t}(k_i) \times \mathcal{S}_{\mu_t}^c(k_i)} \rho_{\pi_{\mu_t}(\mathbf{x},\mathbf{x}')}^{(t)}$
5:      Select $\mathbf{x}_t \in \{\bar{\bar{\mathbf{x}}}_t \triangledown \bar{\bar{\mathbf{x}}}_t', \bar{\bar{\mathbf{x}}}_t \triangle \bar{\bar{\mathbf{x}}}_t', \bar{\bar{\mathbf{x}}}_t \vee \bar{\bar{\mathbf{x}}}_t', \bar{\bar{\mathbf{x}}}_t \wedge \bar{\bar{\mathbf{x}}}_t'\}$.        ▷ Sampling input
6:      $\mathbf{y}_t(\mathcal{D}_t) \leftarrow \mathbf{y}(\mathcal{D}_{t-1}) \cup \{y(\mathbf{x}_t)\}$
7: **end for**
8: Update GP posterior belief: $\{\mu_{T+1}(\mathbf{x})\}_{\mathbf{x} \in \mathcal{X}}, \{\sigma_{T+1}(\mathbf{x})\}_{\mathbf{x} \in \mathcal{X}}$.
9: Construct $\{\mathcal{S}_{\mu_{T+1}}(k_i)\}_{i=1}^m$ as the collection of $m$ top-$k$ sets predicted using $\mu_{T+1}$.
10: **return** $\{\mathcal{S}_{\mu_{T+1}}(k_i)\}_{i=1}^m$.

---

# I   Proof of Theorem 4.4

By choosing $\mathbf{x}_t$ following Lemma 4.3, with probability of at least $1 - \delta$, for all $t \geq 1$,

$$r_{\pi_{\mu_t}(\mathcal{S}_{\mu_t}(k), \mathcal{S}_{\mu_t}^c(k))} \leq \rho_{\pi_{\mu_t}(\mathcal{S}_{\mu_t}(k), \mathcal{S}_{\mu_t}^c(k))}^{(t)} \leq |\mathcal{C}_t(\mathbf{x}_t)| = 2\beta_t^{1/2} \sigma_t(\mathbf{x}_t) \ .$$

Furthermore, from the non-decreasing property of the sequence $(\beta_t)_{t=1}^T$,

$$\sum_{t=1}^T r_{\pi_{\mu_t}(\mathcal{S}_{\mu_t}(k), \mathcal{S}_{\mu_t}^c(k))} \leq \sum_{t=1}^T 2\beta_t^{1/2} \sigma_t(\mathbf{x}_t) \leq 2\beta_T^{1/2} \sum_{t=1}^T \sigma_t(\mathbf{x}_t)$$

By utilizing the result from [5] like Appendix D, we can obtain

$$\sum_{t=1}^T \sigma_t(\mathbf{x}_t) \leq \mathcal{O}(\sqrt{T\gamma_T}) \ .$$

Therefore, with probability of at least $1 - \delta$, for all $T \geq 1$,

$$\sum_{t=1}^T 2\beta_T^{1/2} \sigma_t(\mathbf{x}_t) \leq \mathcal{O}(\sqrt{T\beta_T\gamma_T})$$

i.e.,

$$R_{T,k} \triangleq \sum_{t=1}^T r_{\pi_{\mu_t}(\mathcal{S}_{\mu_t}(k), \mathcal{S}_{\mu_t}^c(k))} \leq \mathcal{O}(\sqrt{T\beta_T\gamma_T}) \ .$$

# J   Active Multiple Set Ordering

The pseudocode of MP algorithm for the active multiple set ordering problem is shown in Algorithm 2. In the rest of this section, we prove the cumulative regret bound of Algorithm 2.

We recall that in (12),

$$(\bar{\bar{\mathbf{x}}}_t, \bar{\bar{\mathbf{x}}}_t') \triangleq \underset{(\mathbf{x},\mathbf{x}') \in \left(\cup_{i=1}^m \mathcal{S}_{\mu_t}(k_i) \times \mathcal{S}_{\mu_t}^c(k_i)\right)}{\arg\max} \rho_{\pi_{\mu_t}(\mathbf{x},\mathbf{x}')}^{(t)}$$

Therefore,

$$\forall i \in \{1, 2, \ldots, m\}, \ \rho_{\pi_{\mu_t}(\bar{\bar{\mathbf{x}}}_t, \bar{\bar{\mathbf{x}}}_t')}^{(t)} \geq \max_{(\mathbf{x},\mathbf{x}') \in \mathcal{S}(k_i) \times \mathcal{S}^c(k_i)} \rho_{\pi_{\mu_t}(\mathbf{x},\mathbf{x}')}^{(t)}$$

$$= \rho_{\pi_{\mu_t}(\mathcal{S}_{\mu_t}(k_i), \mathcal{S}_{\mu_t}^c(k_i))}^{(t)}$$

Furthermore, applying Lemma 3.6 to the input pair $(\bar{\bar{\mathbf{x}}}_t, \bar{\bar{\mathbf{x}}}_t')$, for any $\mathbf{x}_t \in \{\bar{\bar{\mathbf{x}}}_t \triangledown \bar{\bar{\mathbf{x}}}_t', \bar{\bar{\mathbf{x}}}_t \triangle \bar{\bar{\mathbf{x}}}_t', \bar{\bar{\mathbf{x}}}_t \vee \bar{\bar{\mathbf{x}}}_t', \bar{\bar{\mathbf{x}}}_t \wedge \bar{\bar{\mathbf{x}}}_t'\}$

$$|\mathcal{C}_t(\mathbf{x}_t)| \geq \rho_{\pi_{\mu_t}(\bar{\bar{\mathbf{x}}}_t, \bar{\bar{\mathbf{x}}}_t')}^{(t)} \ .$$

Therefore,

$$\forall i \in \{1, 2, \ldots, m\},$$

$$|\mathcal{C}_t(\mathbf{x}_t)| \geq \rho^{(t)}_{\pi_{\mu_t}(\mathcal{S}_{\mu_t}(k_i), \mathcal{S}^c_{\mu_t}(k_i))} .$$

Hence, with probability of at least $1 - \delta$, for all $t \geq 1$,

$$\forall i \in \{1, 2, \ldots, m\},\ r_{\pi_{\mu_t}(\mathcal{S}_{\mu_t}(k_i), \mathcal{S}^c_{\mu_t}(k_i))} \leq \rho^{(t)}_{\pi_{\mu_t}(\mathcal{S}_{\mu_t}(k_i), \mathcal{S}^c_{\mu_t}(k_i))} \leq |\mathcal{C}_t(\mathbf{x}_t)| .$$

As a result, with probability of at least $1 - \delta$, for all $T \geq 1$,

$$\forall i \in \{1, 2, \ldots, m\}, R_{T,k_i} \triangleq \sum_{t=1}^{T} r_{\pi_{\mu_t}(\mathcal{S}_{\mu_t}(k_i), \mathcal{S}^c_{\mu_t}(k_i))} \leq \sum_{t=1}^{T} |\mathcal{C}_t(\mathbf{x}_t)| \leq \mathcal{O}(\sqrt{T \beta_T \gamma_T})$$

where the last inequality comes from Appendix I.

## K   Experiments

All experiments were conducted on a computer equipped with an AMD Ryzen 7 6800HS processor and 16GB of RAM.

To generate a function sampled from a GP, we randomly generate 3 observations $\{(\mathbf{x}_i, y(\mathbf{x}_i))\}_{i=1}^{3}$ and fit a GP model to these observations. Then, we sample function evaluations at all inputs in the domain from the GP posterior distribution. These function evaluations are considered the evaluations of the blackbox function. To generate an observation at a sampling input, we add a Gaussian noise (of $\sigma_n = 0.1$) to the evaluations of the blackbox function at the sampling input.

The expressions for the Branin-Hoo and Goldstein-Price functions are described at [20]. We transform the input domain of these functions to $[0, 1]^2$ and standardize the function evaluations. The input domain is discretized into $n = 100$ points randomly selected in the domain $[0, 1]^2$. The noise is chosen with $\sigma_n = 0.1$.

To perform experiments with the NO$_3$ dataset from Lake Zurich (available at `https://wldb.ilec.or.jp/Lake/EUR-06/datalist`), we standardize the NO$_3$ measurements. Then, a GP model is trained on the standardized dataset to generate the noisy evaluations of the blackbox function over $n = 100$ randomly chosen locations.

We use the logarithmic values of the phosphorus measurements in the soy survey of Brooms Barn [22] to construct a blackbox function. The locations are normalized to the range $[0, 1]^2$ and the logarithmic values of the phosphorus measurements are standardized. Then, we train a GP model to generate the noisy evaluations of the blackbox function over $n = 400$ randomly chosen locations.

To perform experiments with the humidity dataset, we extract the humidity measurements at different locations with the same mote id of 31167 from the Intel Lab data (available at `https://db.csail.mit.edu/labdata/labdata.html`). The humidity measurements are standardized. Then, a GP model is trained to this extracted dataset to generate the noisy evaluations of the blackbox function over $n = 100$ randomly chosen locations.

To remove the potential inefficiency due to repeated sampling in Rand and Var, we provide additional experiments by replacing the Rand and Var baselines with RandNoRepl and VarNoRepl, which do not allow repeated sampling. This modification potentially gives RandNoRepl (random sampling without replacement across different iterations) and VarNoRepl (uncertainty sampling without replacement across different iterations) an additional advantage over our solutions which allow repeated sampling. By avoiding repeated sampling, RandNoRepl and VarNoRepl can sample the input domain more uniformly, whereas our methods might re-sample certain input regions. However, as shown in Figure 5, RandNoRepl and VarNoRepl still do not outperform our solutions. The justification for the efficiency of our solutions is in the nature of noisy observations: With a noise standard deviation of $\sigma_n = 0.1$, a single observation at each input may not suffice to accurately determine the ordering with its neighboring inputs in terms of the function value. Hence, spreading the sampling budget across the whole input domain may not perform well. In contrast, our approach allocates more sampling inputs to the boundary of the top-$k$ set, where it is particularly challenging to check if inputs belong to the top-$k$ set (see Figure 2).

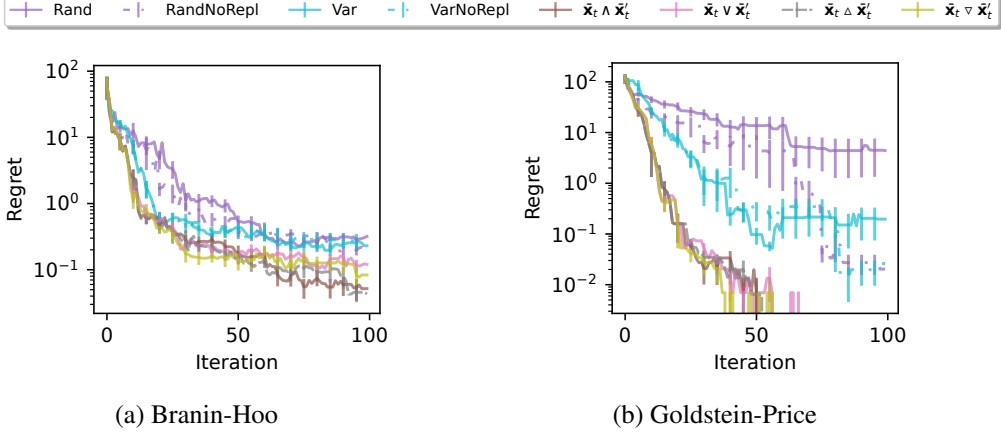

(a) Branin-Hoo  (b) Goldstein-Price

Figure 5: Plots of the regret against the iteration in estimating the top-5 set $\mathcal{S}(t)$.

