# OpenReview forum: "Active Set Ordering"
_NeurIPS.cc/2024/Conference — NeurIPS 2024 poster_

### Official Review · Reviewer_i8Rj · 2024-07-09

**Soundness:** 2
**Presentation:** 3
**Contribution:** 2
**Rating:** 5
**Confidence:** 4

**Summary:**

This submission proposes a novel Mean Prediction (MP) method for active set ordering problem. MP selects pre-defined k inputs with highest Gaussian Process posterior mean values through a novel sampling strategy. Theoretical analysis on the regret, the prediction and the sampling strategy of proposed method is discussed in details, and experiments on synthetic functions and real-life application prove the effectiveness of MP for ordering sets of inputs based on expensive evaluation on black-box function. The submission includes GP-UCB Bayesian optimization as a special case of active set ordering problem.

**Strengths:**

1.  The paper presents adequate theoretical justification for their proposed method. The motivation for studying the active set ordering problem is clearly stated and the proofs on regret bound, prediction and sampling strategy seem valid to me.

2.  The idea of formulating black-box function optimization into set ordering problem presents novelty to some extend. It provides some alternative sampling strategy by proposing the input set $Q_t$ in lines 145-146, which can be more informative than traditional metric such as sampling based on maximizing GP variance.

**Weaknesses:**

1.  My first concern regards the dimensionality of the problem setting. The theoretical justification of MP seems valid for any value of $d$ where $d$ is the dimension of the black-box function. However, in the Experiments section all the problems are 2-dimensional, which is insufficient to show how practical the proposed MP method could be in applications.

2.  This sampling-based method only works on discrete space problems to me. The authors use classic GP model on continuous space as the surrogate model to compute posterior mean, but then sample on the discretized space during inference. This could be one of the major limitations of MP method because a lot of probabilistic information has been missed during the discretization.

**Questions:**

1.  I'm not very convinced by the experiment design. For example the experiment on Goldstein-Price function (lines 248-251), the input domain is discretized into 100 points and a top-5 set based on posterior mean is sampled at each of the 100 iterations. Even repeated sampling is allowed in this paper, I still feel like almost the whole space which only consists 100 points has been sampled by doing so. How do the authors justify the efficiency compared to brute force sampling over the whole sample space?

2.  In all experiment settings, the 100/400 points that represent the input domain $\mathbf{X}$ are randomly sampled from normalized domain $[0, 1]^2$. If I understand correctly, each $\mathbf{X}$ of the 15 repeated experiments is different, which lead to 15 different $S_{\mu_t}(5)$ at the end of iterations. How do the authors decide which one of the $S_{\mu_t}(5)$ is the true top-5 set for the objective black-box problem? (e.g. which 5 locations are the top-5 with highest $NO_3$ concentration in Lake Zurich?)

3.  Could the authors explain how $S_{\mu_t}(k)$ is constructed in line 3 of Algorithm 1? Are the posterior mean of all the points in $\mathbf{X}$ evaluated? Similar for $(\mathbf{\bar{x}_t}, \mathbf{\bar{x}'_t})$ (line 4), are all possible pairs in $S_{\mu_t}(k)$ evaluated?

**Limitations:**

See above for limitations.

---

> ### Author Rebuttal · Authors · 2024-08-07
>
> We would like to thank the reviewer for taking the time to review our paper and for acknowledging the theoretical justification, motivation, and sampling strategy. We will now address the remaining concerns as follows.
>
> > 1. My first concern regards the dimensionality of the problem setting.
>
> We have conducted additional experiments using the Hartmann-6D function to demonstrate the empirical performance of our algorithm in higher-dimensional spaces. Specifically, the input domain consists of $1000$ points, and the input dimension is $6$.
> The results are shown in the attached PDF of the global response.
> They show that our methods consistently outperform other baselines in identifying the top-$50$ set (illustrated in Figure 1b) and in simultaneously finding the maximizer, the top-$100$, and the top-$200$ sets (an active multiple set ordering problem, as illustrated in Figure 1c).
>
> Furthermore, we would like to highlight the practical applications of our problem. As discussed in the introduction, one such application is environmental monitoring, where the input domain is typically two-dimensional, like a geographical area. This focus aligns with several experiments based on real-world datasets presented in our paper.
>
>
> > 2. This sampling-based method only works on discrete space problems to me... This could be one of the major limitations of MP method because a lot of probabilistic information has been missed during the discretization.
>
> Our assumption is that the input domain is a discrete space, and we use an exact Gaussian Process (GP) model to represent the black-box function. According to the marginalization property of GPs, the function values at any finite subset of inputs (including the discrete input domain) follow a multivariate Gaussian distribution. Since we focus solely on function evaluations within this discrete domain and our observations are acquired exclusively from this discrete domain, we believe that no probabilistic information has been lost.
>
> Therefore, we would appreciate any guidance the reviewer can provide regarding the probabilistic information that might have been overlooked.
>
> > 1. I'm not very convinced by the experiment design... How do the authors justify the efficiency compared to brute force sampling over the whole sample space?
>
> Thank you for your careful observation regarding the Goldstein-Price experiment (lines 248-251).
>
> We have revised the experiment by replacing the Rand and Var baselines with RandNoRepl and VarNoRepl, which do not allow repeated sampling. The results are included in the attached PDF file of the global response. This modification potentially gives RandNoRepl (random sampling without replacement across different iterations) and VarNoRepl (uncertainty sampling without replacement across different iterations) an additional advantage over our solutions, which allow repeated sampling. By avoiding repeated sampling, RandNoRepl and VarNoRepl can sample the input domain more uniformly, whereas our methods might re-sample certain input regions.
> However, as shown in Figure 1a of the attached PDF, RandNoRepl and VarNoRepl still do not outperform our solutions.
>
> The justification for the efficiency of our solutions is in the nature of noisy observations: With a noise standard deviation of $\sigma_n = 0.1$, a single observation at each input may not suffice to accurately determine the ordering with its neighboring inputs in terms of the function value. Hence, spreading the sampling budget across the whole input domain may not perform well. In contrast, our approach allocates more sampling inputs to the boundary of the top-$k$ set, where it is particularly challenging to check if inputs belong to the top-$k$ set.
>
> This rationale is also evident in the animation video included in the supplementary materials. Although the uncertainty sampling in the video allows repeated sampling, it distributes samples across the input domain quite evenly (as its goal is to reduce uncertainty throughout the entire input domain).
>
>
> > 2. How do the authors decide which one of the $S_{\mu_t}(5)$ is the true top-5 set for the objective black-box problem?
>
> For each repeated experiment, the set $\\mathcal{S}\_{\\mu\_t}(5)$ and the true top-5 set $\\mathcal{S}(5)$ are computed independently from those in other experiments. The regret for each experiment is calculated using only its own $\\mathcal{S}\_{\\mu\_t}(5)$ and $\\mathcal{S}(5)$. Therefore, it is unnecessary to identify a single set $\\mathcal{S}(5)$ for all repeated experiments, as the regret for each one is computed independently. The plot displays the average and standard error of the regrets across all repeated experiments.
>
> > 3. Could the authors explain how $S_{\mu_t}(k)$ is constructed in line 3 of Algorithm 1?
>
> To construct $\mathcal{S}_{\mu_t}(k)$, the posterior mean values of all inputs in $\mathcal{X}$ are evaluated in $\mathcal{O}(n m_t^2)$ time (time complexity for GP prediction), where $m_t$ is the number of observations at iteration $t$. Subsequently, it takes $\mathcal{O}(n \log k)$ time to identify the top $k$ inputs by using a max heap of size $k$ and scanning through the GP posterior mean of all inputs. It is worth noting that evaluating the posterior mean (and variance) of all inputs in $\mathcal{X}$ is often necessary when $\mathcal{X}$ is finite, as demonstrated in [8] and in cases of finite input domains described in [1,18].
>
> To find $(\\bar{\\mathbf{x}}\_t, \\bar{\\mathbf{x}}'\_t)$ (Equation 11), we iterate through elements in $\\mathcal{S}\_{\\mu\_t}(k) \\times \\mathcal{S}^c\_{\\mu\_t}(k)$, which takes $\\mathcal{O}(k (n - k))$ time (linear in $n$).
>
> ---
>
> Thank you for patiently reading our response. We sincerely hope that the above clarifications address your concerns regarding the algorithm and our experimental results, and hence, improving your opinion on our paper. We will thoroughly incorporate your valuable feedback, along with the additional experiments, into the revised paper.

---

> > ### Comment · Reviewer_i8Rj · 2024-08-13
> >
> > Thank the authors for the response. RandNoRepl and VarNoRepl experiments are interesting and they do perform well given 100 iterations as I expected. By performing the additional comparison to them I'm more convinced by seeing how the proposed MP method close up the regret in fewer iterations for the Goldstein-Price case.
> >
> > This is an interesting idea in general. I'll raise my score to 5.
> >
> > If the authors have time, I'm interested in a comparison between RandNoRepl, VarNoRepl and MP methods on case where both MP and other baselines took all iterations budget to close up the regret in original setting. (e.g. compare RandNoRepl, VarNoRepl and MP methods on Branin-Hoo S(5) with 100 iterations)

---

> > > ### Author Response · Authors · 2024-08-13
> > > **Thank You for Your Reconsideration and Score Enhancement**
> > >
> > > Thank you very much for reviewing our response and for appreciating our additional experimental results, as well as for improving the score. We are glad to hear that our new baseline designs align with your suggestions. If time allows, we hope to address any remaining questions. Accordingly, we have conducted experiments with RandNoRepl and VarNoRepl on Branin-Hoo $\mathcal{S}(5)$ with $100$ iterations.
> > > The results are presented below (together with the results of Rand, Var, and MP in the plot in the paper for reference).
> > >
> > > | Iteration   | 20         | 40         | 60         | 80         | 100        |
> > > |-------------|------------|------------|------------|------------|------------|
> > > | Rand        | 4.80+-1.49 | 1.11+-0.23 | 0.34+-0.07 | 0.25+-0.06 | 0.32+-0.04 |
> > > | RandNoRepl  | 1.68+-0.39 | 0.58+-0.14 | 0.43+-0.08 | 0.32+-0.05 | 0.20+-0.03 |
> > > | Var         | 0.64+-0.11 | 0.34+-0.07 | 0.34+-0.07 | 0.23+-0.05 | 0.23+-0.03 |
> > > | VarNoRepl   | 0.64+-0.11 | 0.37+-0.07 | 0.39+-0.10 | 0.29+-0.08 | 0.29+-0.08 |
> > > | $\bar{\mathbf{x}}_t \wedge \bar{\mathbf{x}}_t'$    | 0.46+-0.07 | 0.29+-0.06 | 0.17+-0.05 | **0.07+-0.02** | 0.05+-0.02 |
> > > | $\bar{\mathbf{x}}_t \vee \bar{\mathbf{x}}_t'$ | 0.56+-0.10 | 0.19+-0.04 | 0.17+-0.05 | 0.11+-0.03 | 0.12+-0.03 |
> > > | $\bar{\mathbf{x}}_t\ \vartriangle\ \bar{\mathbf{x}}_t'$   | **0.42+-0.07** | 0.23+-0.05 | **0.14+-0.04** | 0.09+-0.03 | **0.04+-0.01** |
> > > | $\bar{\mathbf{x}}_t\ \triangledown\ \bar{\mathbf{x}}_t'$   | 0.44+-0.09 | **0.16+-0.03** | 0.18+-0.05 | 0.13+-0.03 | 0.08+-0.02 |
> > >
> > > We observe that RandNoRepl significantly outperforms Rand. However, VarNoRepl performs similarly to Var, as Var is designed to reduce uncertainty across the entire input domain. Due to this characteristic, Var tends not to sample the same input repeatedly when many inputs remain unsampled. Consequently, Var and VarNoRepl behave similarly in the early stages (similar regret). Only when a sufficient number of observations have been made does Var, due to correlations among evaluated inputs, begin to select repeated samples and diverge from VarNoRepl.
> > >
> > > However, since Rand, RandNoRepl, Var, and VarNoRepl are not specifically designed to target the top-$5$ set, our solutions continue to outperform them.
> > >
> > > We forgot to mention that in both the Branin and Goldstein experiments, all algorithms were initialized with a set of $3$ observations. As a result, in the final $3$ iterations, RandNoRepl and VarNoRepl were required to select repeated samples.
> > >
> > > We will incorporate these baselines and the discussion into the revised paper, and we sincerely hope this will address any remaining questions you may have.

---

> > > > ### Comment · Reviewer_i8Rj · 2024-08-13
> > > >
> > > > Thank you for the prompt response, the results are very interesting.

---

> > > > > ### Author Response · Authors · 2024-08-13
> > > > >
> > > > > We're glad to hear that you found our results very interesting. Thank you once again for reviewing our paper. If you have any further questions or insights, please feel free to share them, and we will be happy to address them during the remaining time of the discussion period.

---

### Official Review · Reviewer_Qpjm · 2024-07-11

**Soundness:** 3
**Presentation:** 2
**Contribution:** 2
**Rating:** 6
**Confidence:** 3

**Summary:**

The paper poses a novel active learning problem formulation of active set ordering, in which we aim to identify the data points that yield the top- and bottom-$k$ values via a given objective function.
This active learning goal serves as a compromise between Bayesian experimental design which focuses solely on learning and Bayesian optimization targeting optimization, and poses as an alternative to level-set estimation, especially in scenarios where a level-set threshold is not easily determined.
The authors first define an appropriate metric of regret, propose using the mean prediction of the surrogate Gaussian process to generate an ordering of the data points to recommend to the user, and finally develop an acquisition function that aims to reduce an approximation of the regret resulting from that posterior predictive ordering.
The paper presents various theoretical results bounding the regret of the proposed algorithm, and experiments are conducted to illustrate the empirical effectiveness of the algorithm.

**Strengths:**

I find active set ordering to be an interesting active learning problem that is related to other common active learning problems (experimental design, level-set estimation, Bayesian optimization).
The various algorithmic choices made throughout the paper are reasonable and well-motivated by theoretical insights.
The experiments include a wide range of synthetic and real-world tasks, and do a good job showing that the proposed strategy yields competitive performance against baselines.

**Weaknesses:**

- The presentation of Sections 3 and 4 is a bit hard to follow.
I understand Section 3 serves as the base case where we develop the core tools which are then further extended in Section 4, but the narrative has a jumping-back-and-forth feel that I find somewhat confusing.
- The claims the authors make in Section 1 to motivate the problem might be too strong.
I would say something along the lines of, this work presents an alternative to LSE when setting the threshold is difficult (as opposed to that it should replace it, since sometimes a desirable threshold is known, in which case LSE should be preferred).
- I would include LSE acquisition functions (e.g., STRADDLE) among the baselines in the experiments.

**Questions:**

- The search space sizes in the experiments are quite small.
Does the algorithm scale well to large search spaces?
What’s the computational complexity?
Does it take quadratic time with respect to search space size (because we need to iterate over all pairs)?
- Any insights on how the different variants behave and which should be preferred in which situations?

**Limitations:**

Yes.

---

> ### Author Rebuttal · Authors · 2024-08-07
>
> We would like to thank the reviewer for dedicating their time to review our paper and for acknowledging the interesting research problem situated at the intersection of experimental design, level set estimation, and Bayesian optimization. We are delighted to learn that the reviewer finds our paper to be theoretically sound, well-motivated, and supported by a wide range of experimental results. We would like to address and clarify the remaining concerns as follows.
>
> > I would include LSE acquisition functions (e.g., STRADDLE) among the baselines in the experiments.
>
> As LSE requires a known (or implicit) threshold, which is absent in our problem setting, including their acquisition functions (e.g., STRADDLE) among the baselines in the experiments is not immediately obvious to us.
>
>
> > The search space sizes in the experiments are quite small. Does the algorithm scale well to large search spaces? What’s the computational complexity? Does it take quadratic time with respect to search space size (because we need to iterate over all pairs)?
>
> Updating the GP posterior belief incurs $\mathcal{O}(m_t^3 + n m_t^2)$ computational complexity (including $\mathcal{O}(m_t^3)$ for training and $\mathcal{O}(n m_t^2)$ for prediction, where $m_t$ is the number of observations at iteration $t$ as we use the exact GP model). This complexity can also be reduced using sparse GP approximation. Given the GP posterior belief, our algorithm involves the following major steps outlined in Algorithm 1:
>
> + Line 3: Constructing $\mathcal{S}_{\mu_t}(k)$ takes $\mathcal{O}(n \log k)$ to find the top-$k$ inputs by using a max heap of size $k$ and scanning through the GP posterior mean of all inputs.
>
> + Line 4: We need to scan through the elements in $\\mathcal{S}\_{\\mu\_t}(k) \\times \\mathcal{S}^c\_{\\mu\_t}(k)$, so it takes $\\mathcal{O}(k (n - k))$.
>
> Therefore, the runtime of each iteration is $\mathcal{O}(m_t^3 + n m_t^2 + n \log k + k (n - k))$ which is not quadratic in the search space size ($n$). Specifically, it is linear in $n$.
>
> To further strengthen our experimental results, we have included additional experiments featuring a larger search space, $|\mathcal{X}| = 1000$, and an increased input dimension, $d=6$. They can be found in the attached PDF file of the general response.
>
> > Any insights on how the different variants behave and which should be preferred in which situations?
>
> Based on the cumulative regret bound, we do not have a preference for any particular variant since they all result in the same sublinear cumulative regret bound. However, when $k = 1$, we prefer $\\bar{x}\_t \\triangledown \\bar{x}'\_t$ because it is equivalent to $\\text{arg}\\max\_{x \\in \\mathcal{X}} u\_t(x)$, which can be computed in $\\mathcal{O}(n)$ time without the need to find the pair $(\\bar{x}\_t, \\bar{x}'\_t)$ (Remark 4.5).
> When $k = n - 1$, we prefer $\\bar{x}\_t \\vartriangle \\bar{x}'\_t$ because it is equivalent to $\\text{arg}\\min\_{x \\in \\mathcal{X}} l\_t(x)$, which can also be computed in $\\mathcal{O}(n)$ time without the necessity of identifying the pair $(\\bar{x}\_t, \\bar{x}'\_t)$ (Remark 4.5).
>
> ---
>
> We will revise the paper to incorporate the additional comments on the flow between Sections 3 and 4 as well as the motivation of the problem. We appreciate your patience in reading our response and sincerely hope that the above clarifications, along with the additional experimental results, address your remaining concerns and enhance your perspective on our paper.

---

> > ### Comment · Reviewer_Qpjm · 2024-08-12
> >
> > Thank you for your rebuttal.
> >
> > Regarding LSE, what some do in practice is, for example, to set the threshold to be at some reasonable quantile of the observed data. My goal is to see how much improvement in terms of regret your proposed method leads to compared to naively using LSE in that way in your setting.
> >
> > Overall, I will keep my score.

---

> ### Author Response · Authors · 2024-08-12
> **Thank You and Further Clarification Regarding LSE**
>
> Thank you for your response and for the score in support of our submission. We are eager to address any concerns you may have about our paper, so we would like to provide more details on
> 1. Why comparing an LSE solution with our proposed solutions in the experiments is not straightforward.
> 2. While LSE and finding the top-$k$ set are not empirically comparable, our theoretical analysis suggests that our solutions are as efficient as an LSE approach, even though the top-$k$ problem presents greater computational challenges.
> 3. Our solutions are preferred in the scenario you mentioned, specifically, when setting the threshold as a quantile of the observed data, compared to LSE.
>
> ---
>
> **Empirical comparison**
>
> We agree with the reviewer that in practice, the threshold can be set as a reasonable quantile of the observed data. However, in our problem setting, we are only provided with the size of the top-$k$ set (i.e., the number $k$) without any information about the threshold. This lack of threshold knowledge is one of the key motivations behind our proposed problem, as highlighted in the introduction (lines 25-26):
>
> > "However, without domain knowledge of the black-box function, it is easy to set a threshold that leads to undesirably large or small level sets."
>
> Therefore, running LSE with only this information (i.e., $k$) is not feasible.
>
> If we were to apply LSE with a threshold based on a quantile, the resulting superlevel set would differ from the top-$k$ set in our problem, making a direct comparison between the two methods invalid. This is because estimating a larger set often requires more samples than estimating a smaller set empirically. Additionally, even if we were to carefully select a threshold so that the superlevel set matches the top-$k$ set, the comparison would still be unfair, as the threshold is computed using the knowledge of the true top-$k$ set that is assumed to be unknown in our problem setting and should not be utilized by any solution.
>
> ---
>
> **Theoretical Implication**
>
> While LSE and finding the top-$k$ set are not empirically comparable for these reasons, our theoretical discussion on the upper bound of cumulative regret suggests that our solutions are as efficient as the LSE approach in [8], which is an improved version of STRADDLE.
> This is particularly noteworthy since our problem may be inherently more challenging than LSE. For example, if the black-box function is known, identifying the level set given a threshold can be done by simply comparing all function evaluations against the threshold, which requires a time complexity of $\mathcal{O}(n)$, where $n$ is the size of the input domain. However, finding the top-$k$ set (for $k > 1$) involves using a heap data structure, resulting in a time complexity of $\mathcal{O}(n \log k)$, which is more computationally demanding.
>
> ---
>
> **Motivation for Our Solutions over LSE**
>
> Finally, we would like to emphasize that your observation about the practical use of LSE, where the threshold is chosen as a reasonable quantile of the observed data, precisely highlights the need for our work.
>
> For simplicity, consider a scenario where the input domain consists of 100 points, and the desired level set corresponds to the third quartile (i.e., 25\% of the data points have function evaluations at or above this threshold). One could set $k = 100 \times 0.25 = 25$ and use our solutions to discover the top-$25$ set. The boundary of this top-$25$ set should correspond to the level set they wish to find, without the need to estimate the threshold (such as the third quartile).
>
> In contrast, if one were to use LSE, they would need to estimate the threshold as the third quartile from the observed data. This estimation could be inaccurate in practice due to noise in the observation and a limited number of initial observations. Therefore, even aside from sample efficiency, our proposed solutions are more desirable than LSE in such scenarios.

---

### Official Review · Reviewer_2jNL · 2024-07-12

**Soundness:** 3
**Presentation:** 2
**Contribution:** 2
**Rating:** 6
**Confidence:** 2

**Summary:**

This paper generalizes the best $k$-arm identification problem to Gaussian processes with the goal to estimate the set of the best $k$ function evaluations $f(x)$ on a finite domain $X$, where $k=1$ corresponds to standard Bayesian optimization. The proposed regret notion is a natural adaptation of the common regret $f(x*)-f(x_t)$ in Gaussian process bandits.

The paper is accompanied by some first experiments on benchmark datasets.

**Strengths:**

Interesting problem.

---- rebuttal ----
changed from 5 to 6.

**Weaknesses:**

Finiteness assumption seems rather strong and limits this work. Also, the significance of the results (Thm 3.7 and 4.4) is not fully clear to me, as the paper seems to be mostly relying on known techniques (for kernelized bandits, and Gaussian process bandits), while the related work is not sufficiently well discussed (see questions and limitations).

Am happy to raise my score, if my concerns are addressed. Specifically about the comparison to previous and related literature.

The definition of $\pi_*(X_0,X_1)$ seems only partial, i.e., what if there exists some $x_0$ that are larger than some $x_1$ and some $x_0'$ that are smaller than some $x_1'$

Minor comments:
* The notation is a bit convoluted. E.g., do you really need double bars and dashes for $x$. Even just tilde&dash is not great. E.g., in the beginning of section 3 you could just do $x,x'$ without the tilde (or you use $y$ etc.).
* Please do not mix definitions (in particular of important notation) and lemmas (e.g., Lemma 3.6, 3.1, ...).

**Questions:**

What is the computational complexity/runtime of your approach? Is the runtime exponential in $k$?

How does your work relate to Gaussian bandit papers with correlated/dependent arms (e.g., Gupta et al. 2021, Pandrey et al 2007)

How does the results here relate to e.g., the "Interactive submodular bandit" by Chen et al. [ICML, 2017] (and related), where a submodular function is maximized in a similar fashion (in a GP-style context). Can your $k$ best selection problem be cast as a submodular (or even simply modular) set optimization problem? Are the two regret variants comparable?

**Limitations:**

The finiteness assumption on $X$ seems rather restrictive. Can $X$ be countable, or even say a compact subset / interval? Standard work on Gaussian processes with regret bounds (Srinivas et al., etc.) does not have this restrictions. For finite $X$ less assumptions might be possible, see e.g., Theorem 1 by Krause & Ong (NeurIPS 2011) (for example, no explicit assumptions on the rkhs norm as you do in Lemma 2.2, which they only require for arbitrary domains $X$).

Also in general the related work is not discussed sufficiently well, see questions.

Please, also see the questions.

---

> ### Author Rebuttal · Authors · 2024-08-07
>
> We would like to thank the reviewer for taking the time to review our paper and for appreciating our interesting research problem. We will address the questions and concerns raised by the reviewer as follows.
>
> > the significance of the results is not fully clear to me, as the paper seems to be mostly relying on known techniques
>
> Theorems 3.7 and 4.4 are to show that the cumulative regret of our proposed solution is sublinear. This is a desirable property for BO solutions ([18]) ensuring that the simple regret approaches 0. While the techniques are mainly drawn from the BO literature, it is noted that
>
> + Many existing BO works also employ similar techniques to demonstrate sublinear cumulative regret, with their novelty often lying in new problems. Similarly, our paper makes a unique contribution by addressing the problem of active (multiple) set ordering.
>
> + While we build on established techniques, our paper works with a new notion of regret (pairwise and set ordering) that differs from those in the existing BO literature.
>
> + To the best of our knowledge, we are the first to investigate BO from this pairwise ordering perspective.
>
> > Is the runtime exponential in $k$?
>
> In line 3 of Algo 1, it takes $\mathcal{O}(n \log k)$ to find the top-$k$ inputs by using a max heap of size $k$ and scanning through the GP posterior mean of all inputs. In line 4 of Algo 1, we need to scan through the elements in $\\mathcal{S}\_{\\mu\_t}(k) \\times \\mathcal{S}^c\_{\\mu\_t}(k)$, so it takes $\\mathcal{O}(k (n - k))$. Together with the updating of GP, the runtime of an iteration is $\mathcal{O}(m_t^3 + n m_t^2 + n \log k + k (n - k))$, which is not exponential in $k$. Specifically, it is linear in $k$ as the last term is bounded by $k(n-k) < k n$. Due to lack of space, please refer to our response to reviewer Qpjm for a detailed derivation of the complexity.
>
> > How does the results here relate to e.g., the "Interactive submodular bandit" by Chen et al. [ICML, 2017]?
>
> Our results are different from those in Chen et al. (ICML, 2017):
>
> + First, Chen et al. (2017) model the utility function as *a set function* that takes a set of inputs. Hence, the submodular property is utilized to avoid the exponential nature of the combinatorial problem. In our problem, the utility function (the black-box function) is *defined for each individual input* in the domain, which is equivalent to an additive set function. Our goal is to find the set of $k$ inputs with the highest function evaluations. This is simpler than the general submodular set function, as it *does not incur exponential time complexity for exact solution*.
>
> + Second, the set of interest $S_j$ (using the notation of Chen et al. 2017) is *built sequentially through interaction*, meaning $S_j$ is the set of all sampling inputs up to that point. However, in our problem setting, the top-$k$ set is estimated as the set of top $k$ inputs with the highest GP posterior mean, denoted as $\mathcal{S}_{\mu_t}(k)$. The inputs in this set are *not necessarily those that have been sampled*.
>
> + Third, the optimal set of interest $S^*_j$ is defined as the set having a cardinality of *at most* $T_j$ elements, where *$T_j$ depends on the sampling procedure* (refer to Equation (2) in Chen et al., 2017). This differs from from our optimal (and estimated) top-$k$ set which has a cardinality of *exactly* $k$ where *$k$ is specified before the sampling procedure*.
>
> Therefore, the notion of regret used by Chen et al. (2017) relies on a different concept of optimal set that does not apply to our context. Additionally, it is also noted that the regret definition in Chen et al. (2017) is not based on pairwise ordering, which is the building block of our regret.
>
> > How does your work relate to Gaussian bandit papers with correlated/dependent arms
>
> We assume that Gupta et al. 2021 refers to the paper titled "A unified approach to translate classical bandit algorithms to the structured bandit setting" and Pandrey et al. 2007 refers to the paper titled "Multi-armed bandit problem with dependent arms". These two papers are different from our work as follows.
>
> + The problem: Both Gupta et al. (2021) and Pandrey et al. (2007) are interested in the arm with the maximum reward, while our work focuses on the top-$k$ set.
>
> + Model of dependent arms: Gupta et al. (2021) assumes parametric models for the mean rewards, while Pandrey et al. (2007) assumes that the arms are grouped into known clusters and that the rewards of arms are described by a known generative model with unknown param. In contrast, our work uses a GP (non-parametric) without any assumptions about input clusters.
>
> Given the rich literature, we selected the most relevant works in the kernelized bandit and BO literature (modeling the dependency with GP) [1, 5, 8, 14, 18]. Still, our work differs from them: *the goal of finding top-$k$ set(s)*, *the regret definition based on pairwise ordering*, and *the prediction based on GP posterior mean*.
>
> > The finiteness assumption on $X$ seems rather restrictive.
>
> + The top-$k$ set is undefined for continuous domain. E.g., take $f(x) = -x^2$ with $x \in [-1,1]$. While $x = 0$ is the maximizer, pinpointing the input with the 2nd highest function value is impossible (i.e., defining a top-$2$ set is problematic). Thus, finiteness is critical for solving the problem of finding the top-$k$ set.
>
> + The finiteness assumption is also common in environment monitoring applications (our motivation) such as in [8].
>
> > assumptions on the rkhs norm ... only require for arbitrary domains $X$
>
> Thank you for the insightful comment regarding the RKHS norm. We will revise the paper to include this additional consideration as an alternative.
>
> ---
> Thank you for taking the time to read our response. We sincerely hope that the explanations provided above address your concerns and enhance your perception of our paper. We will incorporate your additional suggestions regarding the notation into the revised paper.

---

> > ### Comment · Reviewer_2jNL · 2024-08-12
> >
> > Thank you for the detailed reply. You are also absolutely right, that the problem is not well-defined in the continuous case. I raised my score.

---

> > > ### Author Response · Authors · 2024-08-12
> > > **Thank You for Reconsidering Our Submission and Raising the Score**
> > >
> > > We sincerely appreciate your reconsideration of our work and the increase in the score. We will carefully incorporate your valuable suggestions into our revisions to enhance the quality of our work. In the meantime, we remain open and eager to provide any further clarifications you may need.

---

### Official Review · Reviewer_gNpn · 2024-07-13

**Soundness:** 3
**Presentation:** 3
**Contribution:** 2
**Rating:** 5
**Confidence:** 4

**Summary:**

This paper introduces the "active set ordering" problem, which aims at recovering the top-k actions in a set by strategically sampling actions. The authors formally define the problem in the regret minimization setting and propose an algorithm for that. They authors upper bound the regret and run experiments the test the proposed algorithm.

**Strengths:**

The authors introduced the problem of recovering top-k actions in the regret minimization setting, and developed an algorithm for that. The authors theoretically show their algorithm enjoys a \sqrt{T} type of regret (after ignoring some problem dependent quantities). Experimental results show the proposed algorithm has good performance.

**Weaknesses:**

1. The authors didn't explicitly quantify some problem-dependent quantities in their regret bound, e.g., \gamma_T and \beta_T. How large these quantities are in different settings?
2. A lower bound analysis is missing, which make it even harder to know if the upper bound is tight or not.
3. While I understand the studied setting is slightly different from top-k arm identification, the proposed algorithm is actually quite similar to algorithms proposed in top-k arm identification: I believe some appropriate acknowledgement is missing in the paper.
4. The wording "set ordering" in misleading: the goal of this paper is not to recover the set ordering but simply identifying the top-k actions in the regret minimization setting.

**Questions:**

See comments above.

---

> ### Author Rebuttal · Authors · 2024-08-07
>
> We would like to thank the reviewer for dedicating their time and effort to reviewing our paper and for recognizing the regret analysis and the experimental results. Additionally, we would like to draw your attention to two other contributions: multiple top-$k$ sets, and the new perspective of ordering for BO in the global response. We will address your remaining concerns as follows.
>
> > 1. The authors didn't explicitly quantify some problem-dependent quantities in their regret bound.
>
> We follow most of the works in the BO and LSE literature (e.g., [8,18]) to demonstrate a desirable asymptotic property of the algorithm: sublinear cumulative regret, i.e., $\lim_{T \rightarrow \infty} R_T / T = 0$. It implies the convergence of the algorithm as $\\min\_{t \\le T} r\_{ \\pi\_{\\mu\_t}( \\mathcal{S}\_{\\mu\_t}(k), \\mathcal{S}^c\_{\\mu\_t}(k) ) } \\le R\_T / T$, i.e., vanishing per-round regret.
>
> In this line of work (from [18]), the cumulative regret bound is often expressed in terms of problem-independent quantities $\beta_T$ and $\gamma_T$, which depend on the kernel.
> The work of [18] discusses the values of $\gamma_T$ for several common kernels (which are referred to in lines 151 and 200). We will clearly state the value of $\gamma_T$ in the revised paper. For example, $\gamma_T = \mathcal{O}((\log T)^{d+1})$ for the squared exponential (SE) kernel (this kernel is used in our paper). The value of $\beta_T$ is elaborated in Lemma 2.2. Hence, by substituting $\gamma_T$ and $\beta_T$ into the cumulative regret bound and simplifying, we can obtain $R_T \le \mathcal{O}^*(\sqrt{T (\log T)^{2d}})$ (where $\mathcal{O}^*(\cdot)$ denotes asymptotic expressions up to dimension-independent logarithmic factors and $d$ is the dimension of the input).
> This is the same as the cumulative regret bound of GP-UCB that is known to match the lower bound of the Bayesian optimization (BO) problem for the SE kernel. We will rely on this result to discuss how tight our cumulative regret bound is relative to the lower bound of the active set ordering problem in the following paragraphs.
>
> > 2. A lower bound analysis is missing.
>
> Let the lower bound of the active set ordering problem be the lower bound of the cumulative regret of the worst-case problem instance over all possible values of $k$. Then it should be at least as large as the lower bound of the special case where $k = 1$. This special active set ordering problem with $k=1$ is the Bayesian optimization (BO) problem, according to our Remarks 4.1 and 4.5. Furthermore, BO has known lower bounds for several common kernels: for example, for the SE kernel, the lower bound of the cumulative regret is $\Omega(\sqrt{T(\log T)^{d/2}})$, (from the paper "Lower bounds on regret for noisy Gaussian process bandit optimization" by Scarlett et al., 2018). Hence, the lower bound of the active set ordering problem is at least $\Omega(\sqrt{T(\log T)^{d/2}})$. Therefore, for the SE kernel, the cumulative regret bound $R_T \le \mathcal{O}^*(\sqrt{T (\log T)^{2d}})$ of our algorithm matches the lower bound up to the replacement of $d/2$ by $2d + O(1)$.
>
> We have discussed the lower bound of the problem by considering the worst-case problem instance over all values of $k \ge 1$, rather than the lower bound of the problem for a specific value of $k \ge 1$. Thus, one may wonder if finding the top-$k$ set for a specific value of $k > 1$ is easier than solving it for $k = 1$ (BO problem). Regarding this question, we suspect that finding the top-$k$ set in active set ordering for a specific value of $k > 1$ is at least as hard as finding the top-$1$ set, i.e., the Bayesian optimization (BO) problem, for the following reasons.
>
> + By intuition, suppose the $f$ is known. Finding the top-$k$ set requires $\mathcal{O}(n\log k)$ time complexity, while finding the top-$1$ set only requires $\mathcal{O}(n)$ time complexity where $n = |\mathcal{X}|$ is the size of the input domain.
>
>
> + Moreover, BO is reducible to the problem of finding a top-$k$ set for any $k \ge 1$. Let us consider a value of $k > 1$ and a BO problem instance defined by a function $f(x)$ where $x$ belongs to a finite input domain $\\mathcal{X}\_f$. Suppose we know an upper bound $U$ of $f(x)$, i.e., $U > \\max\_{x \\in \\mathcal{X}\_f} f(x)$ (this upper bound need not be strict).
> We can create a new function $g$ defined on $\\mathcal{X}\_f \\cup \\mathcal{X}\_g$ where $|\\mathcal{X}\_g| = k-1$ such that:
>     + For all $x \in \mathcal{X}_f$, $g(x) = f(x)$
>     + For all $x' \\in \\mathcal{X}\_g$, $g(x') = U > \\max\_{x \\in \\mathcal{X}\_f} f(x)$.
>
> Then, both the maximizer $x\_*$ of $f$ and $\mathcal{X}\_g$ are part of the top-$k$ set $\\mathcal{S}(k)$ of $g$.
>
> As $|\mathcal{X}\_g| = k - 1$, we have $\\{x\_*\\} = \\mathcal{S}(k) \\setminus \\mathcal{X}\_g$.
>
> Therefore, the problem of maximizing $f$ is reducible to the problem of finding a top-$k$ set of $g$, i.e., finding the top-$k$ set is as least as difficult as solving BO. Hence, the lower bound of the active set ordering problem is at least that of BO problem.
>
> > 3. I believe some appropriate acknowledgement (of top-k arm identification) is missing in the paper.
>
> As you may have noticed, we discussed the difference between our work and the best-$k$ arm identification problem in *the footnote* on page 2. We also acknowledged the similarity with best-$k$ arm identification [12] in the intuition of choosing the input pair in *line 185*. Yet, our approach is different in several ways:
>
> + The choice of sampling input (Lemma 3.6).
> + The regret bound based on the maximum information gain.
> + The justification of mean prediction (Sec. 3.2).
> + Multiple top-$k$ sets (Remark 4.6).
> + A new perspective on the well-known BO solution.
>
> ---
>
> Thank you for your patience in reading our response. We sincerely hope that the above clarifications have improved your opinion of our paper. We will carefully incorporate your valuable feedback and the points discussed above into the revised paper.

---

> > ### Comment · Reviewer_gNpn · 2024-08-09
> >
> > Thank you for your responses. I have increased my score to 5.

---

> > > ### Author Response · Authors · 2024-08-10
> > > **Thank You for Your Reconsideration and Improved Score**
> > >
> > > Thank you very much for your reconsideration and the improved score. We sincerely hope that any previous concerns have been satisfactorily addressed, as we did not identify any remaining issues in your latest comment. We truly appreciate the discussion and will carefully integrate your valuable feedback into the revised paper. Please do not hesitate to reach out if you require any further clarification until the end of the discussion period.

---

### Author Rebuttal · Authors · 2024-08-07

We sincerely thank all the reviewers for their time and effort in reading and evaluating our paper. We are encouraged by the positive feedback and appreciation of our work regarding its novelty (reviewers Qpjm and i8Rj), theoretical soundness (reviewers Qpjm and i8Rj), and experimental results (reviewers gNpn and Qpjm).

---

We are pleased to summarize our contributions as follows.

+ We address the problem of estimating the top-$k$ set of a black-box function modeled by a Gaussian process. Specifically, we propose novel regret notions of pairwise ordering and set ordering. Then, we analyse the regret to justify our prediction and sampling strategy.

+ We extend our solution to accommodate the estimation of multiple top-$k$ sets, as discussed in Remark 4.6 and Section 5.2, driven by practical motivation in environmental monitoring applications.

+ We offer a novel perspective on the well-known GP-UCB algorithm through the lens of ordering (that has not been previously explored in Bayesian Optimization literature), which yields several nuanced insights (Remark 4.5).

---

In our response to the reviewers, we have provided additional clarifications, including:

+ Discussion on the lower bound of the proposed problem (reviewer gNpn).

+ Comparison between our work and existing studies (reviewer 2jNL).


+ The linear time complexity in the input domain size and linear time complexity in $k$ (reviewers 2jNL and Qpjm).

+ Additional experiments involving a 6-dimensional input domain in the PDF file attached in this response (reviewer i8Rj).

---

We genuinely hope that our response effectively addresses the reviewers' concerns and enhances their opinion of our paper. Your feedback is invaluable to us, and we are eager to incorporate it to further improve our work.

---

### Author Response · Authors · 2024-08-14
**Thanks to All Reviewers and the Area Chair**

We would like to thank all reviewers for your time, effort, and valuable feedback in reviewing our work, and to the Area Chair for coordinating the process. We greatly appreciate your patience in considering our responses and for your decision to raise the scores and maintain a supportive evaluation. It is our sincere hope that we have adequately addressed your concerns and questions regarding our paper. We will carefully incorporate your comments and suggestions in our revision.

---

### Decision · Program_Chairs · 2024-09-25

**Decision:**

Accept (poster)

**Comment:**

The reviewers generally agreed that this was an interesting problem, and the algorithmic tools were solid. The top-k problem clearly has lots of applications. While some reviewers were lukewarm about the work, we had a followup discussion with them and generally all reviewers were ok with accepting the paper for NeurIPS. I want to commend the authors for doing a good job in the discussion. But I want to recommend that particular care is taking in preparing the camera-ready, especially in incorporating the feedback received throughout this review process.